

# Seasonal variation of tropospheric bromine monoxide over the Rann of Kutch salt marsh seen from space

C. Hörmann[1], H. Sihler[1,2], S. Beirle[1], M. J. M. Penning de Vries[1], U. Platt[2], and T. Wagner[1]

[1]Max-Planck-Institute for Chemistry (MPI-C), Mainz, Germany
[2]Institute for Environmental Physics, University of Heidelberg, Heidelberg, Germany

*Correspondence to:* C. Hörmann (c.hoermann@mpic.de)

**Abstract.** The Rann of Kutch (India/Pakistan) is one of the largest salt deserts in the world. Being a so-called 'seasonal salt marsh', it is regularly flooded during the Indian Summer Monsoon. We present 10 years of bromine monoxide (BrO) satellite observations by the Ozone Monitoring Instrument (OMI) over the Great and Little Rann of Kutch. OMI spectra were analysed using Differential Optical Absorption Spectroscopy (DOAS) and revealed recurring high BrO VCDs up to $1.4 \times 10^{14}$

molec/cm$^2$ during April/May, but no significantly enhanced column densities during the monsoon season (June–September). In the following winter months, the BrO VCDs are again slightly enhanced while the salty surface dries up. We investigate a possible correlation of enhanced reactive bromine concentrations with different meteorological parameters and find a strong relationship between incident UV radiation and the total BrO abundance. In contrast, the second Global Ozone Monitoring Instrument (GOME-2) shows about four times lower BrO VCDs over the Rann of Kutch than found by OMI and no clear

seasonal cycle is observed. One reason for this finding might be the earlier local overpass time of GOME-2 compared to OMI (around 9:30 vs. 13:30 LT), as the ambient conditions significantly differ for both satellite instruments at the time of the measurements. Further possible reasons are discussed and mainly attributed to instrumental issues. OMI additionally confirms the presence of enhanced BrO concentrations over the Dead Sea valley (Israel/Jordan), as suggested by former ground-based observations. The measurements indicate that the Rann of Kutch salt marsh is probably one of the strongest natural point

sources of reactive bromine compounds outside the polar regions and is therefore supposed to have an significant impact on local and regional ozone chemistry.

## 1 Introduction

Reactive halogen species (RHS) are well known to play an important role in atmospheric chemistry of both the troposphere and stratosphere (e.g. Platt and Janssen, 1995; Saiz-Lopez and Glasow, 2012, and references therein). For almost 30 years, ground-

based observations in polar regions, at active volcanoes, within the mid-latitude marine boundary layer (MBL) and over salt lakes indicate the potential of these species to significantly influence the oxidation capacity of the troposphere, possibly on a global scale, via the catalysed depletion of ozone ($O_3$). After the so-called ozone depletion events (ODEs) were first observed in the 1980s by local measurements in the Arctic (Oltmans, 1981; Oltmans and Komhyr, 1986; Barrie et al., 1988), especially reactive bromine compounds were identified to be responsible for the ozone destruction. In particular, a strong anti-correlation





of the ozone concentration with filterable bromine (Barrie et al., 1988, 1989) was found. In later studies, the involvement of bromine chemistry was directly confirmed by additional measurements of bromine monoxide (BrO) using the Differential Optical Absorption Spectroscopy (DOAS) technique (Platt and Stutz, 2008), with BrO mixing ratios of up to 17 ppt (Hausmann and Platt, 1994). Subsequent observations showed even higher mixing ratios of up to 40 ppt (e.g. Avallone, 2003; Frieß, 2004;

Hönninger et al., 2004; Pöhler et al., 2010; Peterson et al., 2015, and references therein). In the meantime, an overall picture of the BrO horizontal distribution was obtained by observations from satellite instruments in the late 1990s (Wagner and Platt, 1998; Richter et al., 1998; Chance, 1998). It became clear that the ODEs can be associated with huge BrO 'clouds' of several thousands of km² extent.

During the last three decades, BrO has also extensively been measured in the MBL by ground-based DOAS observations, but

the observations revealed relatively low BrO mixing ratios of 1–10 ppt (e.g. Leser et al., 2003; Saiz-Lopez, 2004; Read et al., 2008; Martin et al., 2009). Volcanic BrO has not only been observed by a large number of ground-based DOAS observations at quiescent degassing volcanoes since the early 2000s (e.g. Bobrowski et al., 2003; Oppenheimer et al., 2006; Bobrowski and Platt, 2007; Boichu et al., 2011; Bobrowski and Giuffrida, 2012; Lübcke et al., 2014), but also during minor and major volcanic eruptions by satellite instruments (Theys et al., 2009a; Rix et al., 2012; Hörmann et al., 2013).

## 15  1.1  BrO observations over salt lakes

In contrast to BrO observations in other regions, reports about reactive halogen species observations over salt lakes were only infrequently published. Following the first ground-based DOAS observations of BrO over the Dead Sea with peak mixing ratios of 86 ppt by Hebestreit et al. (1999), later studies were able to confirm the results and found even higher mixing ratios of 220 ppt maximum (e.g. Matveev et al., 2001; Tas et al., 2005), representing the highest BrO mixing ratios ever observed in the

20 atmosphere (outside volcanic plumes). The vertical distribution of BrO above the Dead Sea and the corresponding dynamics were more recently discussed in Holla et al. (2015). Much lower mixing ratios of up to 6 ppt were found over the Great Salt Lake (United States) by Stutz et al. (2002) as well as more than 20 ppt at Salar de Uyuni (Bolivia) by Hönninger et al. (2004). Further field campaigns were conducted by Holla (2012) in Namibia (Walfish Bay), Botswana (Sua Pan), South Russia (El'Ton and Baskuntschak), Mauretania (Sebkha N'Dramcha), South West Australia (Lake Stubbs, Lake Orr, Lake King, Lake Tay and

25 Lake Chlorine) and Cape Verde (including artificial solar salt ponds at Santa Maria and the Pedra Lume caldera, which is filled with oceanic water). However, none of these measurements showed significantly enhanced BrO or iodine oxide (typically less than 2 ppt for MAX-DOAS measurements). A model study by Tas et al. (2006) focusing on the Dead Sea emphasized the possibly important role of aerosols for the release of reactive bromine, while another study by Smoydzin and von Glasow (2009) suggested an additional direct bromine release from sea water. The autocatalytic reaction cycle that is associated with the

30 rapid production of BrO (so-called 'bromine explosion', Platt and Lehrer, 1997) could recently be observed under laboratory conditions within a smog chamber (Buxmann et al., 2012). In the latter study, environmental conditions like salt composition, pH value, temperature and relative humidity (RH) were varied and a BrO build-up above a model salt pan could be studied in detail, indicating a strong dependency on RH - possibly controlled by the thickness of resulting water microlayers on the salt



crust. However, the detailed chemical mechanism still remain unclear as corresponding model calculations were not able to reproduce the findings.

## 1.2 The Rann of Kutch

The Rann of Kutch is a 'seasonal' salt marsh, stretching from the Indo-Pakistani border into the Kutch District of India's largest state, Gujarat. With more than 30000 km$^2$ it is the largest salt desert in the world and additionally one of the hottest areas of India with summer temperatures around 50°C and winter temperatures decreasing below 0°C during night. The Rann (the name originates from the Hindi word for *desert*) can be subdivided into a large northern part ('Great Rann of Kutch') and the considerably smaller 'Little Rann of Kutch' (Figure 1), located at the south-eastern border of Gujarat. During India's summer monsoon (June/July - September/October), the flat desert of salty clay and mudflats (which lies, on average, 15 meters a.s.l) is flooded by tidal water as well as freshwater from nearby rivers and standing water from extensive rainfall (Figure 2). Due to these rather extreme climatical conditions, the area is widely uninhabited. After the monsoon season, the accumulated shallow water layer evaporates, leaving large parts of the Rann covered by a snow white salt crust.

Large industrial evaporation ponds are located near the coast of the Great and Little Rann of Kutch (indicated by red marked areas in Figure 2a, left) and used for the production of salt and the subsequent recovery of elemental bromine from sea bittern, i.e. mother liquor left after recovery of common sodium chloride from brine (Mehta et al., 2003). Especially all over the Little Rann, the salt is also harvested from hand-built salt evaporation ponds that are constructed by thousands of local families year after year (Pacha, 2013), after having been destroyed by the annually recurring monsoon flood (see also Figure 1–3 of the Supplementary Material).

Atmospheric measurements by ground-based instruments have not been performed in the Rann of Kutch yet, probably due to the difficult climatic conditions. Satellite observations, however, provide the unique possibility to investigate the entire area remotely over a long time period.

## 2 Instruments

### 2.1 Ozone Monitoring Instrument

The Ozone Monitoring Instrument (OMI) is part of NASA's Earth Observation System (EOS) program and is carried by the Aura satellite, launched in July 2004 into a sun-synchronous polar orbit at 705 km altitude (Levelt et al., 2006). Aura crosses the equator at about 13:30 local time. The three channel UV/Vis push-broom spectrometer covers a wavelength range from 270–500 nm at a moderate spectral resolution of 0.42–0.63 nm. Featuring a two-dimensional CCD detector with a near nadir ground pixel size of 13×24 km$^2$ (up to 28×150 km$^2$ at the swath edge) and a large swath width of 2600 km, the measurements achieved daily global coverage until the first appearance of the so-called 'row anomaly' in June 2007. Today, global coverage is achieved after two days, as almost half of the OMI ground pixels are affected by the 'row anomaly' (KNMI, 2015). So far,





OMI has been successfully used to monitor enhanced BrO vertical column densities (VCDs) during polar spring, associated with the ODEs (e.g. Salawitch et al., 2010; Choi et al., 2012) and during volcanic eruptions (Theys et al., 2014).

## 2.2 Global Ozone Monitoring Experiment-2

The second generation of the Global Ozone Monitoring Experiment (GOME-2) is a series of three identical instruments that
are part of the MetOp satellite program operated by the European Organisation for the Exploitation of Meteorological Satellites (EUMETSAT). The first GOME–2 instrument, GOME-2A, is carried by MetOp-A and was launched into a sun-synchronous polar orbit at 800 km altitude in October 2006 (Callies et al., 2000). The second instrument (GOME-2B, carried by MetOp-B) was launched in September 2012, GOME-2C will be carried by the MetOp-C satellite to be launched in 2018. MetOp-A crosses the equator at about 9:30 local time. In this study, only GOME-2A observations are analysed.
The GOME-2A instrument is a 4 channel UV/Vis grating spectrometer that observes the Earth's atmosphere in near–nadir viewing geometry. Originally, a scanning mirror provided a cross-track swath width of 1920 km (at observation angles up to $50°$ off-nadir) resulting in global coverage within 1.5 days (EUMETSAT, 2005; Munro et al., 2006). GOME-2A measures both the radiance of sunlight reflected by the Earth's atmosphere and the solar irradiance, covering the wavelength region of 240–790 nm at moderate spectral resolution of 0.2–0.4 nm. With a nadir pixel size of $40 \times 80$ km$^2$, GOME-2A observed 4
15  times smaller ground pixels than its predecessor GOME on ERS-2, but 10 times larger when compared to OMI ground pixels (see Section 2.1). After several test configurations in early 2013, the swath-width of GOME-2A was reduced to 960 km (nadir pixel size of $40 \times 40$ km$^2$) to achieve daily global coverage via the so-called 'tandem operation' mode together with GOME-2B, which covers the remaining daily gaps. Like OMI, GOME-2(A) data have been successfully used to observe enhanced tropospheric BrO VCDs during polar spring (e.g. Begoin et al., 2010; Theys et al., 2011; Sihler et al., 2012) and during
volcanic eruptions worldwide (Theys et al., 2009a; Hörmann et al., 2013).

## 3 BrO DOAS retrieval

The data of both satellite instruments were analysed at MPI-C for BrO column densities by using the Differential Optical Absorption Spectroscopy (DOAS) technique (Platt and Stutz, 2008). The GOME-2 data was evaluated using the algorithm developed by Sihler et al. (2012). The OMI BrO retrieval follows the general settings of the GOME-2 retrieval: a wavelength
range from 336–360 nm was used, covering 4 adjacent absorption bands. In addition to the BrO cross section from Wilmouth et al. (1999), ozone cross sections at 223 and 243 K (Gür et al., 2005), O$_4$ (Greenblatt et al., 1997), NO$_2$ (Vandaele et al., 2002), OClO (Bogumil et al., 2003) and SO$_2$ (Bogumil et al., 2003) were used. Furthermore, two Ring spectra (one calculated for a sun reference spectrum, the other one using the former scaled by $\lambda^4$), an inverse reference spectrum and a 5th order polynomial were included in the BrO retrieval.
OMI data affected by the 'row anomaly' (first appearing after June 25th, 2007) were excluded from the retrieval (KNMI, 2015). Due to differences in the measurement sensitivity of single CCD detector pixels, OMI BrO column density maps suffer from cross-track striping, which is especially apparent for weak absorbers like BrO. To overcome the striping effect, mean geomet-





rical BrO VCDs (appropriate for an stratospheric absorber) calculated for each pixel row individually over a remote area over the Pacific ($\pm20°$N, 105–175°W) were subtracted from all daily measurements. Since the contribution of stratospheric BrO to the total column is hereby removed as well, the median BrO column across the track was re-added. Please note that the BrO VCDs finally used in this study are corrected for the stratospheric contribution to the total column by a specifically local

separation (see Sect. 3.2).

The measured spectra are additionally affected by permanent and transient hot pixels, leading to increased noise at certain wavelengths for individual CCD detector rows. Such erroneous measurements can be easily identified by abnormally large fitting residuals at affected wavelength positions and again lead to strongly elevated BrO VCD stripes. Following the suggestions by Chance (2007), intensities measured at pixels showing very strong residual discrepancies ($>4\sigma$) from an initial fit were

10 excluded for a final second fit. The vast majority of former suspiciously anomalous BrO slant column densities (SCDs) showed realistic columns similar to nearby measurements after the correction.

### 3.1 Clouds

The observation of tropospheric trace gases is significantly affected by radiative transfer due to cloud coverage in several ways. If the trace gas is located above a thick cloud layer, satellite measurements might show an increased sensitivity. On the

15 other hand, clouds may completely shield trace gases close to the surface. To minimize the influence of clouds, only OMI measurements with an effective cloud fraction of less than 30% were considered. For this purpose two operational Level 2 OMI cloud products are provided by NASA:

1. OMCLDO2, for cloud detection using the $O_2$–$O_2$ absorption near 477 nm (e.g. Acarreta and de Haan, 2002; Vasilkov et al., 2008)

2. OMCLDRR, using information of the so-called filling-in of solar Fraunhofer lines caused by rotational-Raman (RR) scattering in the atmosphere (Ring effect) within the 346–354 nm spectral range (e.g. Joiner et al., 2002; Stammes et al., 2008)

Although the OMCLDO2 algorithm is most commonly used, it turned out that the algorithm almost always mis-classifies the bright surface of the salt marsh as cloud and is thereby unsuitable for the BrO analysis presented here. In contrast, the

25 OMCLDRR algorithm seems to better distinguish between the bright surface and clouds, probably because of an increased contrast in the UV in comparison to the visible wavelength range that is used in the OMCLDO2 algorithm. It should, however, be noted that a reanalysis of the data applying a lower cloud filter threshold of only 20% showed that the OMCLDRR algorithm sometimes misinterprets the bright surface of the salt marsh as clouds as well. This is in particular the case during the first 3–4 months after the Rann is flooded (October/November–December/January). As the remaining water evaporates, a very clean and

30 bright surface remains as indicated by increased reflectivity; this effect can be seen in MODIS true color images as in Figure 2. In order to provide statistically relevant monthly averaged BrO VCDs, only grid pixels that were covered at least ten times by the daily measurements were finally taken into account. It can, however, not be completely ruled out that some of the actual cloud free measurements are sorted out using the cloud filter.





For GOME-2, the only available cloud detection algorithms are the operational FRESCO (Koelemeijer and Stammes, 2001; Koelemeijer et al., 2002) and ROCINN (Loyola R., 2004; Rozanov et al., 2006) products, which both use wide parts of the oxygen A-band spectrum to determine the effective cloud fraction. Similar to the OMI OMCLDO2 algorithm, both FRESCO and ROCINN fail to distinguish between the bright surface of the salt marsh and clouds. Therefore only a qualitative comparison

of the unfiltered cloudy data will be presented in Section 4.5.

### 3.2   Local stratospheric background correction

Because all satellite measurements include the total atmospheric BrO column, the data have to be corrected for the stratospheric fraction to retrieve the tropospheric BrO column over the salt marsh area. The stratospheric BrO distribution varies little with latitude and even less with longitude (Theys et al., 2009b). Therefore, BrO VCDs determined assuming a geometrical airmass

factor were corrected by subtracting the results of a two-dimensional spatial polynomial fit of 3rd degree $n$ applied to the daily measurements as described in Hörmann et al. (2013):

$$V^*_{\text{trop,i}} \approx V^*_{\text{tot,i}} - \sum_{m,n=0}^{3} V^*_{\text{strat,i}} \times x_i^m \times y_i^n \qquad (1)$$

where $V^*_{\text{strat,i}}$ are the fitted stratospheric BrO VCDs at the centre coordinates $x$ and $y$ [°] of satellite pixel $i$ within a large area around the Rann of Kutch (18–30°N, 62–78°E). To minimize the influence of possibly enhanced BrO VCDs over the salt

marsh, the actual Rann area (22.5-25.5°N, 67.5–72.5°W) was excluded from the polynomial fit of the stratospheric correction approach. The resulting corrected geometrical BrO VCDs ($V^*_{\text{trop}}$) were reconverted into 'tropospheric' SCDs ($S^*_{\text{trop}}$) by multiplication with the geometrical AMF.

### 3.3   Radiative transfer

As the radiative transfer for a tropospheric absorber is not adequately represented by the geometrical approximation, the

'tropospheric' $S^*_{\text{trop}}$ were multiplied with box air-mass factors (box-AMF) calculated by the Monte Carlo radiative transfer model (RTM) McArtim (Deutschmann et al., 2011) to retrieve final BrO VCDs. The simulations were conducted near the strongest BrO absorption band (338.5 nm) assuming cloud-free conditions. Two main model runs were performed for different homogeneous BrO layer profiles (0-400 m, 0-1 km and 0-2 km):

1. No additional aerosols (varying surface albedo of 0.10, 0.15 and 0.20)

2. Additional homogeneous aerosol layer at 0-1 km (single-scattering albedo (SSA): 0.9, asymmetry parameter (AP): 0.72 and varying aerosol optical depth (AOD) of 0.4, 0.7 and 1)

As baseline properties, a surface albedo of 0.15 (based on typical OMI reflectivity values at 331 nm; available via the NASA OMSO2 Level 2 OMI SO$_2$ product and shortly described in Carn et al., 2013), a homogeneous BrO layer at 0-1 km and an AOD of 0.7 were chosen. All baseline aerosol properties assumptions (AOD, SSA and AP) were based on monthly mean



OMI observations at 342.5 nm (via the NASA OMAEROG Level 2 aerosol product, described in Torres et al., 2007). As the OMAEROG product often shows unrealistically high parameter values exclusively over the Rann area (e.g. monthly mean AODs of more than 4), the product seems to have some problems with the exceptionally bright surface of the salty desert. Additionally, the associated parameters seem to be affected by increased cloud coverage during the monsoon season.

In order to estimate the actual aerosol parameters despite these problems, mean values for AOD, SSA and AP close to (but outside of) the Rann area were chosen. An estimate of the uncertainties caused by the a priori assumptions will be given in Section 4.2.

# 4   Results

The individual daily measurements of each month from October 2004 to December 2014 were gridded on a regular lat-lon grid
with a spatial resolution of $0.1°$, from which monthly mean BrO VCD maps were calculated. Figure 3 shows the monthly mean BrO VCDs (10-year-averages for 2004–2014) for baseline RT settings (see Section 3.3). It is clearly visible from the maps that the BrO VCDs slowly rise during the first months of each year and reach a maximum during April/May ($\approx 6 \times 10^{13}$ molec/cm$^2$). In the second half of the year, the values are much lower. During the monsoon (June–September), the enhancement almost completely disappears. The low values during the monsoon are, however, at least partly caused by the shielding effect of
remaining cloud cover over the observed area (as additionally indicated by enhanced background noise in the southeastern part of the shown region due to the reduced statistics). After the monsoon (October–December) the BrO VCDs remain at a low level, reaching magnitudes similar to those found at the beginning of the year. Although the individual years 2004–2014 partly show a less prominent seasonal variation of the absolute BrO VCDs over the salt marsh, the temporal evolution is similar for all analysed years (see Supplementary Material for all time series). The annual mean BrO VCD maxima were mostly found during
April (2006–2009, 2011–2014) and twice in May (2005, 2010), with a 10-year-maximum of about $1.4 \times 10^{14}$ molec/cm$^2$ in April 2009.

    Regarding the Great Rann of Kutch, a closer look to the annual pattern indicates that the BrO VCDs typically first rise in the western part during March, before clearly enhanced VCDs can be seen over the whole area in April and mostly over the middle/eastern part during May (Figure 3, March–May). It remains unclear if this behaviour originates in the different ambient
conditions in different areas and months or if it is caused by transport due to a steady westerly wind (see Section 4.4 and Figure 7). A significant contribution of halogen emissions from the large artificial evaporation ponds that are used for salt/bromine mining is unlikely, as no trend can be seen from the measurements although the industrial facilities were massively expanded after 2009 (see Supplementary Material for MODIS observations of these industrial complexes).

    The maximum BrO VCDs over the Little Rann of Kutch are generally lower compared to the Great Rann, with a maximum of
$6.1 \times 10^{13}$ molec/cm$^2$ in May 2010. Like for the Great Rann, the annual maximum BrO VCDs typically appear during April–May, but remain at a very low level close to the detection limit for all other months (Figure 3). Even during April/May, the maximum BrO VCDs over the Little Rann are about a factor of 3 lower than over the Great Rann.



### 4.1 Variation of surface reflectivity and sensitivity

The strongly enhanced albedo of the salty crust is expected to have a significant effect on the sensitivity for near-surface BrO. This may lead to an apparently enhanced BrO VCDs over the Rann when compared to nearby areas featuring a lower albedo due to a potential enhanced tropospheric BrO background and the underestimation of the corresponding AMF. To investigate

the possible influence of the bright surface on the spatial pattern of BrO VCDs, the reflectivity at 331 nm as seen by OMI (see Section 3.3) was used to calculate monthly averaged maps for the same data that were used for the BrO VCD maps (i.e. using a cloud filter CF<30%). Figure 4 shows the resulting mean reflectivity at 331 nm for 10-years-averaged months during 2004–2014. At first glance, especially the northern Great Rann of Kutch can be clearly identified for all months outside the monsoon season by the strongly increased reflectivity. During the monsoon (June–September) the shape of the salt marsh vanishes as the

area is flooded and more and more (bright) clouds shield the Rann (despite the applied cloud selection criteria). The extensive cloud coverage generally leads to a widely enhanced mean reflectivity over the whole investigated area, especially the Indian mainland. After the monsoon, the mean reflectivity is typically even higher (20–23%) than during the first months of the year (15–20%), probably because the evaporating water leaves a clean, bright and salty crust. This effect can also be recognized in the corresponding MODIS true color images (compare Figure 2). A comparison of Figure 3 and 4 leads to the interesting

finding that the reflectivity is smallest during months that typically show the highest BrO VCDs (April/May). The minimum reflectivity over the Great Rann appears for May 2005–2014 (as well as for most individual years; see Supplementary Material for all time series). Maximum BrO VCDs are found in April 2005–2014. Almost no BrO VCD enhancement can be observed after the monsoon months, whereas the reflectivity is highest. Generally, there is no strong spatial correlation of the BrO VCDs to the mean reflectivity patterns. While there is only little variation in the regional reflectivity within the Rann area over

different months, the spatial distribution of BrO VCDs changes and even seems to progress into the eastern part of the Great Rann from March–May.

In contrast to the Great Rann, the Little Rann cannot clearly be identified from averaged reflectivity maps as corresponding values stay close to surrounding areas ($\approx$7-10%), with the lowest values occurring during May 2005–2014, when enhanced BrO VCDs clearly show up.

### 4.2 Estimation of total BrO mass and mixing ratio

To estimate the total amount of BrO for each month, the BrO VCDs within the Rann area (22.5-25.5°N, 67.5–72.5°E) were summed up and converted into total masses ($M_{BrO}$=95.903 g/mol) by assuming the afore-mentioned baseline properties (i.e. a BrO layer thickness of 1 km; see Section 3.3). Figure 5 shows the resulting BrO mass over the Rann in the time period between October 2004 and December 2014 (blue areas) next to the BrO mass determined from a neighbouring reference area

of the same size to the west of the Rann region (red line). For the sake of clarity, all measurements that fall within the monsoon season (June–September) are indicated in green. Corresponding reflectivity values are shown by the blue dotted line.

The largest BrO amount can be regularly observed in April/May, ranging from 2100t in May 2007 up to 4700t in May 2010. Taking into account all OMI measurements, the mean BrO mass during spring (March–May) is calculated to be (2700±917)t,



where the uncertainty represents the corresponding standard deviation $\sigma$. In contrast, the mean mass over the reference area was found to be about a factor of 13 lower (212±105)t. The deviation from zero can probably be attributed to the imperfect polynomial correction for the stratospheric BrO background. During the monsoon season (June–September) the BrO mass declines over the Rann and remains at a comparatively low level of (1066±753)t, often followed by a local maximum di-

rectly after the monsoon (October) before another minor drop leads to an annual minimum around December. While the mean BrO masses over the Rann during the monsoon are still about 3x larger than over the corresponding reference area for most years (279±138)t, they are associated with a high uncertainty due to the influence of cloud coverage and associated small number of measurements. In this context it should be mentioned that the reference area is generally less affected by (convective/orographic) clouds as large parts are located over the Arabian Sea. During wintertime (November–February), the mean

BrO masses remain at the same low level (1067±560)t for the area over the Rann, (187±131)t over the reference area), however, the uncertainties are now much smaller as the Rann region is only sporadically affected by clouds.

Assuming a standard atmosphere and the baseline properties for the BrO layer (0–1 km), mixing ratios of 20-66 ppt maximum can be roughly estimated from the maximum BrO VCDs during March-May 2005–2014 (mean mixing ratio: 35±10 ppt). In reality, however, the mixing ratios remain highly uncertain as the true atmospheric properties (in particular the BrO vertical

profile and the aerosol amount) are unknown. To point out the strong influence of all presumed parameters for the mixing ratio estimation (i.e. the AMF calculation and the assumption of the BrO layer height), Table 1 shows the results for a variation of all key parameters in comparison to the baseline scenario. While varying aerosol parameters or even the general presence of aerosols (i.e. AOD=0) only lead to small changes (-8 to 6 %) of the estimated BrO mixing ratio, a variation of the surface albedo (-14 to 20 %) and especially the BrO layer thickness (-23 to 34 %) has a rather strong effect on the corresponding

mixing ratios. Overall, the final uncertainties can be estimated to be about -33 to 73 %.

## 4.3   Correlation with meteorological parameters

The influence of meteorological parameters on the 'bromine explosion' reaction cycle is still largely unclear and remains a subject of intensive discussion within the scientific community. Whereas e.g. the role of blowing snow in the activation of bromine over first-year Antarctic sea ice has recently been analysed in situ (Lieb-Lappen and Obbard, 2015), the influence of

environmental conditions like temperature, relative humidity or pH value at salt lakes were almost exclusively investigated by using an artificial salt pan within a smog chamber (Buxmann et al., 2012). It was found that the corresponding BrO mixing ratios were almost one magnitude higher at a relative humidity of 60% when compared to experiments at 37% or 2%. This is probably caused by quasi-liquid water layers on the salty crust that seem to support the activation of reactive bromine.

To investigate possible dependencies of the observed total BrO mass over the Rann of Kutch on humidity and other parameters,

data from the European Centre for Medium-Range Weather Forecasts (ECMWF) for the whole investigated time period and within a 1°× 1° area located over the central part of the Great Rann (23.5–24.5° N, 69–70° E) were used to calculate monthly averaged values for different meteorological parameters around the time of the OMI overpass (9 UTC). Figure 6 shows the annual variation of the BrO mass together with total precipitation (P), cloud coverage (CC), relative humidity (RH), temperature (T), boundary layer height (BLH) and UV radiation at the surface ($\text{UV}_{rad}$). At first glance, the parameters can be divided into





two groups by their seasonal behaviour (indicated by different colours for left and right column in Figure 6a–f). While P, CC and RH are closely related to the Indian monsoon, T, BLH and $UV_{rad}$ are closely related to each other via their dependence on solar irradiation. The latter three parameters correlate very well with the total BrO mass during the first half-year, before a second maximum shows up after the end of the monsoon. Focusing on the first half-year (i.e. the gradual increase of observed

BrO VCDs starting in January until the beginning of the monsoon season in June), a linear fit applied to the BrO mass leads to high correlation coefficients for T ($r^2$=0.87), BLH ($r^2$=0.93) and $UV_{rad}$ ($r^2$=0.85). In contrast, the other parameters show no correlation to the BrO mass as indicated by $r^2$ values of 0.03 (P), 0.05 (CC) and 0.03 (RH).

To further investigate whether a combination of these quantities can be used to model the annual BrO variation over the salt marsh, a systematic multilinear regression analysis was conducted. Starting from a single parameter up to a combination of all

10 six (P, CC, RH, T, BLH and $UV_{rad}$), a simple linear function was determined to describe the observed BrO cycle. The resulting $r^2$ value for each linear fit of the total BrO mass and individual multivariable function was finally used to indicate the goodness of fit. Figure 6g depicts the model results for all functions featuring the highest $r^2$ value for a given number of parameters. Table 2 additionally lists the corresponding fit results and $r^2$ values. For a single variable, a linear function (LF) depending on the incident UV radiation already describes the annual BrO variations fairly well ($r^2$=0.70, purple dashed line in Fig. 6g). As

the UV radiation remains a permanent component of all resulting functions independent of the total number of fitted variables, the photochemical aspect of the 'bromine explosion' is emphasized. Surface temperature appears to have a nearly negligible influence and is only included for the fit function featuring all six parameters ($r^2$=0.97, LF6, red dashed line in Fig. 6g), showing almost identical results when compared to LF5 (including all parameters but temperature). Especially precipitation and relative humidity are clearly needed to fit the low level BrO variations during the monsoon and wintertime as can be seen by LF2/3

(light blue and orange dashed line in Fig. 6g, $r^2$=0.83/0.91). Taking into account four parameters ($UV_{rad}$, CC, RH and BLH) already leads to a linear function (LF4, black dashed line in Fig. 6g) that is well capable of describing all main features of the mean annual BrO observations ($r^2$=0.95). Interestingly (but self-evident), the features that were contributed by accounting for the precipitation appear to be widely included in the CC parameter for LF4, as both parameters are closely related to each other (the second highest $r^2$ value for LF3, 0.86, was achieved by using CC instead of P).

The seasonal cycle of BrO formation over the Rann of Kutch can be described reasonably well as a linear function of meteorological parameters for the 10 years averaged data on a monthly basis. It should, however, be noted that this simple approach fails to adequately reproduce the seasonal cycle when monthly averages from individual years are considered. In particular, the less pronounced BrO maxima in 2007/2008/2011/2012 are clearly underestimated and also the highly variable winter months are not very well captured by the linear functions. It has to be kept in mind that a linear model assumption of independent

variables constitutes a rather simplistic approach to describe the satellite observations and can therefore only be used to get a rough idea about the general circumstances that are needed for the extensive formation of BrO over the salt marsh.

## 4.4 Seasonal wind pattern

Other possible parameters that might (at least indirectly) influence the 'bromine explosion' mechanism and the observed BrO spatial pattern are wind speed and main direction. Figure 7 and 8 (upper panels) show the 10-years-averaged monthly BrO




VCDs over the Rann of Kutch along with the corresponding mean wind pattern that were calculated using daily ECMWF data (9 UTC). The lower panels in Figure 7 and 8 additionally illustrate the wind speed and the frequency of different wind directions. Starting with slow northeastern winds in January ($\approx 2$ m/s), the wind blows into eastern direction during months exhibiting strongly enhanced BrO VCDs (March–May). Towards the monsoon season, the winds get significantly stronger (up to 4 m/s in May) and reach their maximum strength in July with more than 7 m/s during 90% of all days, consistently coming from WSW. During these months, the strong wind speed supports the flooding of the Rann area by tidal water from the Arabian Sea. After the monsoon, the wind speed drops rapidly (1–2 m/s) and the wind turns towards east again. The following winter time is dominated by calm winds from the Northeast.

It is interesting to realize that the distinct seasonal wind patterns over the Rann of Kutch can be linked to different phases of the observed BrO occurrence. Apart from the close to zero BrO VCDs during the monsoon months that are dominated by strong winds from the southwest, the maximum BrO VCDs in March–May are accompanied by slow westerly winds, while the low BrO VCDs during wintertime are influenced by slow winds from the northeast. As mentioned above, the low BrO VCDs during the monsoon are at least partly caused by cloud shielding. Outside the monsoon season, the different wind patterns lead to differences in the effective time that an air mass stays in close contact to the salty crust of the Rann. Because the Great Rann of Kutch expands about 250 km from East to West (but only about 70 km from North to South), single air packages ideally may have remained about four times longer over the Great Rann for westerly wind conditions in April than during December, which is dominated by wind coming from the north. Aerosol particles that are needed for the 'bromine explosion' mechanism might therefore have an increased probability to be swirled up from the salty surface crust in significant amounts during springtime. An examination of the actual aerosol appearance over the Rann remains, however, difficult, as available satellite aerosol products are strongly influenced by the bright surface of the salt marsh (see also Section 3.3).

For the Little Rann of Kutch, the wind direction has a much lower influence on the residence time of the air over the salt marsh, as this part of the salt marsh is much smaller and only slightly stretched in east-western direction ($\approx 50 \times 80$ km).

## 4.5 Comparison of OMI results with GOME-2 observations

The GOME-2 instrument overpasses the Rann of Kutch at about 9:30 local time (and therefore 4 hours before OMI). To investigate the diurnal evolution of BrO VCDs over the Rann area, 5 years of GOME-2 data (2007–2011) were evaluated for BrO in a similar way as for OMI (see Section 3 for details). As mentioned in Section 3.1, no cloud filter could be applied without losing a significant number of probably cloud free GOME-2 observations because the operational cloud products mistakes the bright surface for clouds. Therefore, only a qualitative seasonal intercomparison of the data will be presented in the following: No cloud filter is used and consequently, a *geometrical* AMF is used to calculate BrO VCDs (i.e. the data are only corrected for the viewing geometry; explicit radiative transfer calculations are not involved). Please note that the geometrical AMF is this time also applied to OMI data to guarantee the consistency of both data sets. Because fully cloudy cases are not filtered out, low BrO VCDs are not only due to small near-surface BrO concentrations, but also to cloud shielding (possibly even outside of the monsoon season). However, it should be noted that most days outside the monsoon months (June–September) are only effected by low cloud fractions according to MODIS and ECMWF data. In addition, no differences in cloud cover were found





that could be attributed to the different overflight time of the instruments throughout the year.

As it can be seen from Figure 9 (lower panel), the seasonal mean BrO VCD maps only indicate a very low enhancement over the salt marsh, which is close to background noise level. Please note that the GOME-2 BrO VCDs are presented using the same colorbar as the corresponding OMI results (Figure 9, upper panel) to emphasize the low levels of the GOME-2 BrO VCDs.

During the monsoon season (here July–September, Figure 9, lowermost left panel), neither satellite instrument detects a clear enhancement, as massive cloud coverage shields the salt marsh area, but whereas OMI data clearly show a seasonal cycle with a pronounced maximum in April-June, no obvious seasonal variation can be seen in the GOME-2 data. One explanation for the very small BrO VCDs measured by GOME-2 (compared to OMI) may be differences in chemistry due to different ambient conditions at the early overpass time of GOME-2 (see below). Furthermore, the weak BrO VCD enhancement in the GOME-2

data could at least be partly caused by radiative transfer effects due to the increased surface albedo of the salt marsh. However, as the surface albedo shows a typical seasonal variation in contrast to the GOME-2 BrO VCDs (see Figure 4), this effects seems to be rather small. While radiative transfer simulations suggest a possible overestimation of the BrO VCDs by a factor of 1.5 (50%) due to an inadequate consideration of the bright surface (Section 3.3), this effect can be neglected for OMI, where peak BrO VCDs over the Rann are up to a factor of 16 larger than for areas outside the Rann at low surface albedo.

In principle, there are several instrumental and chemical reasons why the BrO VCDs over the Rann of Kutch derived from GOME-2 are lower than those from OMI:

1.  The spatial resolution of the GOME-2 instrument is $40 \times 80$ km$^2$, while the near nadir ground pixel size of OMI is $13 \times 24$ km$^2$. GOME-2 spectra are therefore generally expected to be less sensitive to localised emission sources, as

larger ground pixels include a larger fraction of sunlight from areas outside the investigated object (the typical extent of the salt marsh area showing clearly enhanced BrO VCDs is less than that of a GOME-2 ground pixel).

2.  GOME-2 is known to suffer from instrumental degradation (especially in the UV wavelength region) starting in 2008/2009 (Dikty and Richter, 2012). A significant increase in the scatter of retrieved VCDs has been observed, particularly for BrO. This, in addition to the generally low signal-to-noise-ratio of GOME-2 in the UV compared to OMI (Fioletov et al.,

2013), has led to a lower sensitivity of the instrument to small BrO concentrations.

3.  One of the main chemical reasons for much lower BrO VCDs using GOME-2 might be that the measurements take place about 4 hours earlier when compared to OMI observations ($\approx$ 9:30 vs. 13:30 LT). At the time of the morning overpass, the 'bromine explosion' mechanism has presumably not progressed very far, as solar irradiance is approximately 50% less than during OMI's afternoon overpass (according to ECMWF data) and the process is photolytically driven. Furthermore,

$O_3$ is needed for the rapid build-up of BrO, which might be more easily available during the morning on the one hand, but may lead to differences in the spatial BrO distribution patterns for GOME-2 when compared to OMI. Observations at other salt lakes have shown that the largest BrO VCDs can be expected close to noontime (e.g. Holla et al., 2015).





4. The boundary layer height during the GOME-2 overflight in April/May is significantly lower (≈2km) than for the OMI measurements (≈3km). As the BLH increases towards noon, BrO originating from the ground might be transported to higher altitudes where it could be more easily detected by the OMI as the instrument's sensitivity generally increases for elevated layers. Additionally, the increasing BLH might lead to an increased mixing-in of tropospheric $O_3$ from higher altitudes and thereby lead to further formation of BrO.

5. Other meteorological parameters (like relative humidity) are expected to influence the efficiency of the 'bromine explosion' (Buxmann et al., 2012). While surface temperature (and associated parameters like e.g. boundary layer height) are lower during the GOME-2 overflight, relative humidity is about 30% higher. However, the detailed role of meteorological conditions on the 'bromine explosion' mechanism remain unclear (see Section 4.3).

Although all of these effects probably contribute to the observed differences between OMI and GOME-2 observations, the reasons for this discrepancies are still a matter of further research.

## 4.6 The Dead Sea (Israel/Jordan)

OMI data were additionally analysed over another salt lake, the Dead Sea, but the results will only be shortly discussed in the following. A more detailed analysis of the data (or for further locations) exceeds the scope of this paper.

As already mentioned in Section 1.1, BrO over the Dead Sea has been frequently observed by ground-based DOAS measurements during recent years. While the results of Hebestreit et al. (1999) suggested that the salt pans over the southern part of the Dead Sea are the main source of the observed BrO, Matveev et al. (2001) and Tas et al. (2005) concluded that BrO is produced over all parts of the sea, although the frequency and intensity of BrO production seemed to be more intense over the southern basin. Figure 10 shows the mean BrO VCDs for summer months with relatively low cloud coverage (April–October) for the complete data set from 2005–2014, including an additional cloud filter of <30%. As it can be seen from the map, the mean BrO VCDs are clearly enhanced over the southern part of the Dead Sea, while only weakly enhanced VCDs can be identified over the northern part. The results seem to confirm the ground-based findings, although the maximum BrO VCDs are shifted towards the southwestern inland, probably because of northeasterly winds that predominate in this area (Matveev et al., 2001). While the Rann of Kutch shows specifically high BrO VCDs in OMI mean maps (at least during April/May), BrO from other salt lakes like the Dead Sea may generally not be identified as easily from the OMI data for several reasons. Besides lower BrO concentrations, DOAS satellite observations of optically weak absorbers like BrO are known to suffer from erroneous evaluation results caused by imperfect correction of the Ring effect (Wagner et al., 2009). This effect appears more prominent if the measured scene contains high elevation and bright surfaces like snow-covered mountain ranges (e.g. Salar de Uyuni in the Andes, Chile) in combination with strong cloud coverage during the year. The OMI BrO analysis of individual months over the Dead Sea reveals that the whole Middle East region is indeed more regularly affected by clouded scenes when compared to the Kutch area. Resulting monthly mean maps were hence significantly affected by low statistics. Furthermore, the analysis is complicated by the mountainous areas along the Jordan and Hula Valley and the associated influence on the correction of the Ring effect. In this context it should be noted that significantly enhanced BrO VCDs were frequently observed north of





the Dead Sea Valley and close to Tripoli, Libya (not shown). As there is no obvious source for reactive bromine species to the authors' knowledge, it remains unclear if these findings are caused by residual structures in the DOAS evaluation or actually caused by unknown bromine sources.

## 5 Conclusions

Satellite measurements have been used to monitor the seasonal cycle of BrO formation over the Rann of Kutch seasonal salt marsh. In particular, we presented the first space-based measurements of BrO originating from a salt marsh/salt lake, emphasizing the capability of satellite instruments to monitor remote and hardly accessible areas for long-term measurements. Up to now, no ground-based atmospheric measurements have been conducted at the Rann of Kutch to the authors' knowledge, despite (or precisely because of) the unique environmental conditions prevailing on-site. The results indicate that the Rann of Kutch salt marsh is probably one of the strongest natural point sources of reactive bromine compounds outside the polar regions and is therefore supposed to have an significant impact on local and regional ozone chemistry.

The OMI measurements reveal a typical annual BrO formation cycle over the Rann, with maximum BrO VCDs appearing during March–May, coinciding with the strongest UV irradiation at the surface. These findings confirms that BrO is not directly emitted from the surface, but driven by photochemistry, which involves the release of BrO precursors like molecular bromine ($Br_2$) as a part of the autocatalytical 'bromine explosion'. During the monsoon (June–September), the whole Rann of Kutch area is flooded, and a massive decrease of the BrO VCDs is observed. This effect is, however, partially caused by increased cloud cover, as shielding of the surface strongly affects satellite measurements of trace gases close to the ground. Only slightly enhanced BrO VCDs show up in wintertime (November–February). No clear influence of the Rann's surface albedo can be seen from OMI UV reflectivity measurements. While enhanced BrO VCDs may appear due to an unconsidered increase of the surface albedo, the Rann's surface appears relatively dark during the time of the maximum BrO VCDs (probably because of first moistening towards the monsoon season) and BrO can be hardly detected during the winter months, when the surface appears brightest.

A first attempt to describe the annual BrO cycle based on a simple linear parametrisation of different meteorological quantities indicates that, in addition to UV irradiation, the variation of the boundary layer height is an essential parameter needed to describe the annual BrO peak, which can be at least partly explained by the higher sensitivity of the satellite for elevated layers. For an adequate description of the near-surface BrO variations, relative humidity and precipitation also play an important role. The seasonal wind conditions can be additionally linked to the observed seasonal variation of the BrO VCDs: The maximum VCDs in March–May are accompanied by westerly winds, which may lead to an increased swirling up of aerosol particles due to the east-west expansion of the Rann and the thereby increased residence time of air masses over the salt crust. In contrast, low VCDs during wintertime are affected by slow winds from the northeast.

Corresponding GOME-2 measurements during the morning (4 hours earlier than OMI) show about 4 times lower BrO VCDs and no clear seasonal cycle, aside from minimum column densities during the strongly cloud affected monsoon months. Although instrumental reasons may contribute to these findings, the results indicate the fundamental influence of ambient meteo-





rological conditions at the time of the satellite overflight (e.g. the UV radiation at the surface is about 50% lower), and that the 'bromine explosion' still needs to evolve during the morning hours in order to allow BrO concentrations to build up to values that may be detected from space.

Additional OMI measurements of enhanced BrO VCDs over the Dead Sea demonstrate the potential of satellite instruments for the global observation of reactive halogen species over salt lakes. By the improved temporal and in particular spatial resolution of upcoming satellite instruments like the TROPOspheric Monitoring Instrument (TROPOMI, Veefkind et al., 2012, launch expected for mid 2016), the number of daily salt lake observations will further increase. Furthermore, satellite instruments on geostationary orbits featuring several measurements per day over the same location like Sentinel-4 (Ingmann et al., 2012, expected for 2019) will allow to investigate the diurnal evolution of the 'bromine explosion' in more detail (not only over salt lakes). The significantly enhanced BrO VCDs over the Rann of Kutch in combination with the unique local conditions on the other hand strongly suggest to undertake a ground-based measurement field campaign in that area to better constrain the general release mechanisms of reactive bromine compounds from salt marshes and lakes.

*Acknowledgements.* We would like to thank the agencies providing the satellite data: The OMI OMSO2 (including reflectivity at 331nm), OMCLDO2 and OMCLDRR products as well as MODIS AQUA/TERRA true color and channel 7-2-1 images are archived and distributed from the Goddard Earth Sciences Data & Information Services center (NASA). We acknowledge EUMETSAT for providing GOME-2A Level 1 data and ECMWF for providing meteorological parameters over India/Pakistan.





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





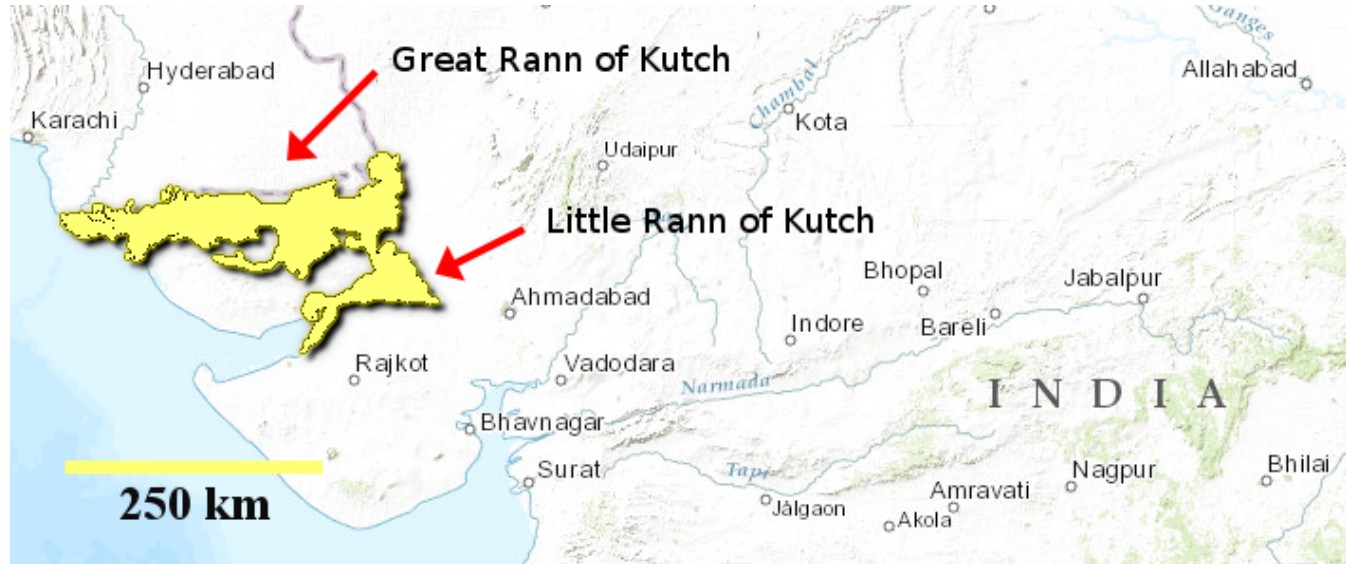

**Figure 1.** Location of the Great and Little Rann of Kutch. While about 10% of the Great Rann belong to Pakistan, the main part as well as the Little Rann are located in the Kutch District of Gujarat, India (adapted from Fund, W.: Rann of Kutch seasonal salt marsh, 2014. Retrieved from http://www.eoearth.org/view/article/155658, last access on 28 January 2016, CC BY-SA 2.5).





**Figure 2.** MODIS true color (TC, left) and Band 7-2-1 images (right) over the Rann of Kutch for selected days in a) February, b) March, c) June, d) September and e) November 2005 (13.2., 24.3. 12.6., 21.9. and 17.11.), illustrating the flood during the monsoon. Water can be identified from the 7-2-1 images as dark blue/black, while clouds and bright surfaces appear light blue. The Rann appears as a bright surface earlier the year, before it darkens due to inflowing water (indicated by yellow arrows in b) and c), right column; compare with the mean reflectivity shown in Figure 4). During monsoon time, the entire Rann is often completely flooded and large areas are dominated by mud (TC image in September). Locations of industrial evaporation ponds are indicated by red marked areas in a).



**Figure 3.** Seasonal variation of monthly mean BrO VCDs with CF<30% over the Rann of Kutch as seen by OMI during 2004-2014. While maximum VCDs are clearly detected in April/May for the Great and Little Rann, only small enhancements can be seen during wintertime. Results during the monsoon season (July–September) include less measurements due to increased cloud frequency leading to a higher noise level, especially in the south-eastern part of the shown region. Please note that OMI data are only available since October 2004, i.e. no data for January–September 2004 are included.



**Figure 4.** Seasonal variation of the monthly mean reflectivity at 331 nm over the Rann of Kutch as seen by OMI during 2004-2014 (same data selection as in Figure 3, i.e. for CF<0.3). Only the Great Rann of Kutch can be clearly identified due to its bright surface compared to the surrounding areas. During the monsoon (July–September), wide areas are still affected by clouds as indicated by the increased background, including the Rann area. It is important to note that enhanced BrO VCDs in Figure 3 appear only at certain parts of the salt marsh (area of enhanced reflectivity) and especially for rather low reflectivity in April/May.




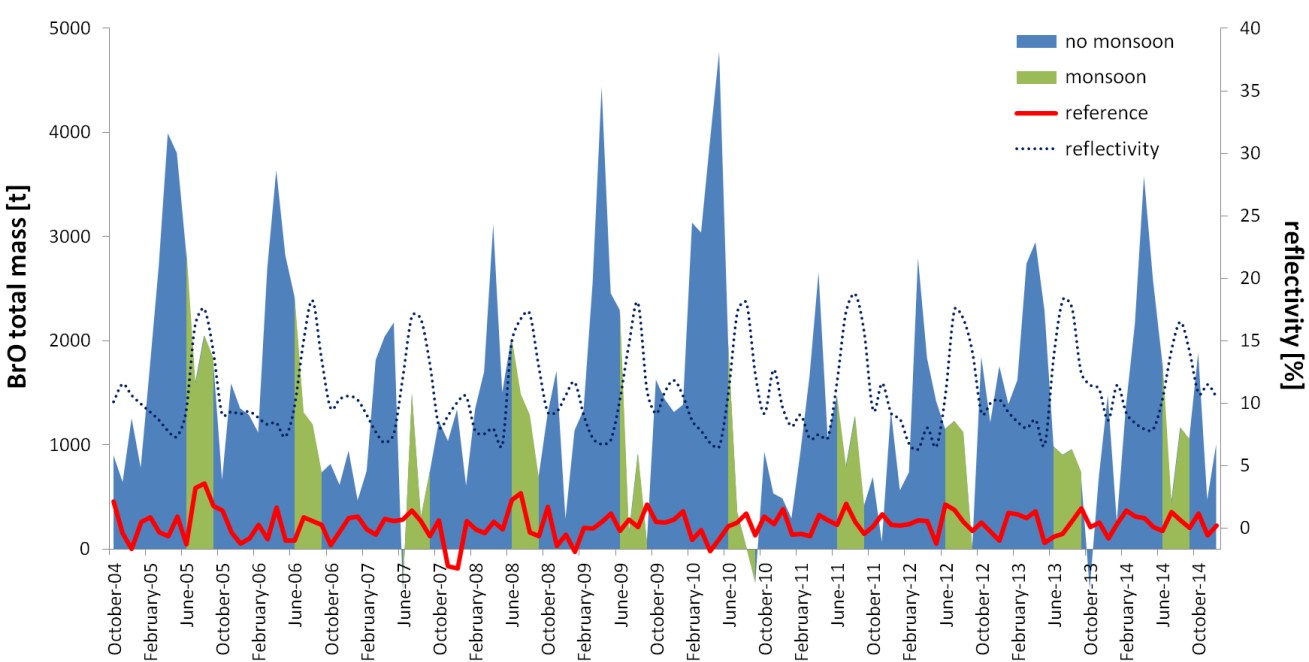

**Figure 5.** Total BrO mass over the Rann of Kutch area calculated from monthly averaged BrO VCD OMI data for the baseline scenario (i.e. a homogeneous layer reaching from the surface up to 1 km) from October 2004 until December 2014. During monsoon months (green) the reflectivity (blue dotted line) shows maximum values due to cloud coverage, while the BrO mass is lowered (partly due to cloud shielding). Maximum BrO masses are typically accompanied by a rather low reflectivity. For comparison, BrO masses obtained over a corresponding reference area of the same size directly west of the considered region are indicated by the red line.





**Figure 6.** Monthly averaged variation of ECMWF meteorological parameters (thin lines) in comparison to the total BrO mass (blue area) over the Rann of Kutch during 2004-2014. The parameters can be divided into two groups by their seasonal behaviour (precipitation, cloud coverage and relative humidity are shown within the left column in green, **a–c**; surface temperature, boundary layer height and UV radiation within the right column in orange, **d–f**). The latter mentioned parameters correlate very well with the BrO mass during the first half-year, before a second maximum shows up after the end of the monsoon. The results of a multilinear regression analysis suggest that a simple linear model (linear functions LF1–LF6, **g**) can be used to adequately describe the annual variations (see text and Table 2 for further details).

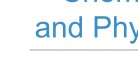

**Figure 7.** Mean wind field at 10 m altitude over the Rann of Kutch for the time around the OMI overpass (9 UTC) during January–June 2005–2014 according to ECMWF data together with the BrO VCDs observed from OMI (upper panel). The area is mainly affected by westerly winds at times when BrO VCDs are high. A shift of the BrO distribution in May can be explained by a turning of the main wind direction (and increasing velocity) towards north-east. Corresponding wind speeds (lower panels) are rather low (1-3 m/s) before they increase with the onset of the monsoon season (>7 m/s)





**Figure 8.** Same as in Figure 7, but for July to December 2004–2014. The area is mainly affected by strong southwestern winds during monsoon months (>7 m/s). After the monsoon, the wind speed rapidly drops ($\approx$2 m/s) and the wind direction changes from west (October) to northeast during wintertime.



**Figure 9.** Seasonal mean BrO VCDs over the Rann of Kutch area for OMI (upper panel) and GOME-2 (lower panel) in 2007-2011: **a)** January–March, **b)** April–June, **c)** July–September, **d)** October–December. Even for the largest BrO VCDs as seen by OMI during April–June, the GOME-2 VCDs are close to the detection limit and might be at least partly explained by retrieval artefacts caused by the bright surface. Note that, in contrast to Figure 3, no cloud filter was applied (see text).





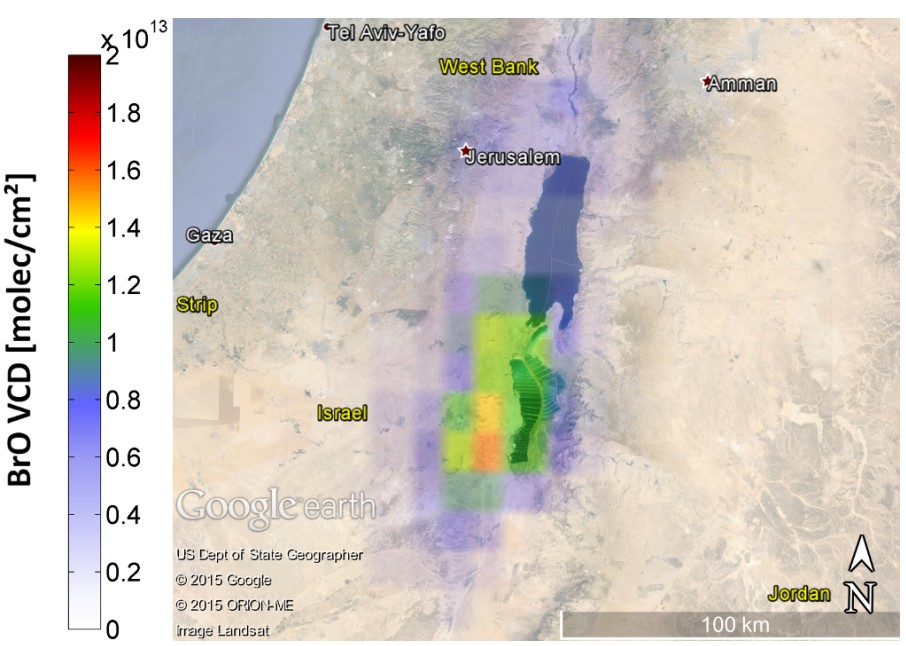

**Figure 10.** Averaged BrO VCD over the Dead Sea as seen by OMI during summer months 2005-2014 (April–October). A clear enhancement of the BrO VCDs can be seen over the southern part of the Dead Sea, confirming former ground-based observations by (Hebestreit et al., 1999), (Matveev et al., 2001) and (Tas et al., 2005).



**Table 1.** Uncertainty of BrO mixing ratio estimation based on the variation of assumed properties (BrO profile, surface albedo, aerosol optical density) for the AMF calculation. (*) AOD=0 corresponds to a non-aerosol scenario.

| a priori | baseline | alternatives | mixing ratio [ppt] | $\Delta$BrO [%] |
|---|---|---|---|---|
| layer profile | 0–1 km | 0–400 m | (47$\pm$13)ppt | +34% |
| | | 0–2 km | (27$\pm$8)ppt | -23% |
| AOD | 0.7 | 0* | (32$\pm$9)ppt | -8% |
| | | 0.4 | (34$\pm$10)ppt | -3% |
| | | 1.0 | (37$\pm$11)ppt | +6% |
| surface albedo | 0.15 | 0.1 | (42$\pm$12)ppt | +20% |
| | | 0.2 | (30$\pm$9)ppt | -14% |

**Table 2.** Best fits from the multilinear regression analysis in order to model the seasonal BrO mass variation over the Rann of Kutch in dependency of UV surface radiation (UV), precipitation (P), cloud coverage (CC), relative humidity (RH), boundary layer height (BLH) and surface temperature (T).

| # | linear function | $r^2$ |
|---|---|---|
| 1 | $m_{BrO} = a_1 \cdot UV$ | 0.71 |
| 2 | $m_{BrO} = a_1 \cdot UV + a_2 \cdot P$ | 0.83 |
| 3 | $m_{BrO} = a_1 \cdot UV + a_2 \cdot P \qquad + a_4 \cdot RH$ | 0.91 |
| 4 | $m_{BrO} = a_1 \cdot UV \qquad + a_3 \cdot CC + a_4 \cdot RH + a_5 \cdot BLH$ | 0.95 |
| 5 | $m_{BrO} = a_1 \cdot UV + a_2 \cdot P + a_3 \cdot CC + a_4 \cdot RH + a_5 \cdot BLH$ | 0.97 |
| 6 | $m_{BrO} = a_1 \cdot UV + a_2 \cdot P + a_3 \cdot CC + a_4 \cdot RH + a_5 \cdot BLH + a_6 \cdot T$ | 0.97 |