# Peer review of "Seasonal variation of tropospheric bromine monoxide over the Rann of Kutch salt marsh seen from space"

_Atmospheric Chemistry and Physics, 2016_

## Referee Comment (RC1) · Anonymous Referee #1 · 16 May 2016

Hörmann et al. report seasonal variations of tropospheric BrO abundances over the Rann of Kutch (salt desert/seasonal lake at the border of India and Pakistan) using BrO tropospheric column retrieved using OMI UV measurement during 2005-2014. This is a first attempt to quantify tropospheric BrO over salt lakes using satellite measurements. This study agrees well with the scope of Atmospheric Chemistry and Physics. I recommend this article to be published in ACP given that the following major and minor concerns are addressed.

Major comments:

1) Total column BrO retrieval using DOAS (Differential Optical Absorption Spectroscopy) has significant uncertainty depending on selection of fitting window, up to

~50% or higher, which may affect the magnitude of tropospheric column BrO and thus the BrO mass abundance quantified using that column. The tropospheric column BrO and BrO mass abundance is as correct as the magnitude of total column BrO retrieved in the given fitting window, and it should be mentioned in the manuscript (in Sect. 3). In addition, comparing the BrO mass abundances from authors' retrieval and those from the OMBRO operational product (using the same approach) will give a good, solid example to show possible uncertainties in mass abundances calculated from satellite BrO measurements.

2) Although stratospheric column BrO has little variations at the region of study, the "magnitude" of assumed stratospheric column BrO can affect the magnitude of tropospheric column BrO and thus the magnitude of mass abundances of tropospheric BrO. Such uncertainties in the magnitude are particularly important in this study, since this study aims "quantification" of BrO mass abundances instead of merely tracking the seasonal and temporal variations. How does the estimate of stratospheric column BrO compare with the other estimates using models, such as Theys et al. 2009b and Salawitch et al., 2011? How much can the estimated tropospheric BrO abundances change along with the assumed loadings of stratospheric column BrO? These points need to be addressed in the manuscript (probably in Sect. 3.2).

3) Sect 4.5: Authors used a geometrical AMF for tropospheric BrO over the Kutch of Rann using GOME-2 data. However, I do not agree with that the BrO column retrieved using nadir-viewing UV measurements and a geometrical AMF over not-so-bright surfaces (albedo ~ 0.15) has the capability to distinguish tropospheric BrO contribution. If authors cannot prove that using a geometrical AMF has such capability or provide the GOME-2 BrO analysis using tropospheric AMF from reasonable radiative transfer calculation, the entire section need to be removed.

Minor comments:

1) Latitude and longitude of the Rann of Kutch need to be specified in the early part of

the manuscript for readers who are not familiar with the area of study.

2) Page 2 line 6: "an overall picture of the BrO horizontal distribution" need to be "an overall picture of BrO horizontal distribution in the troposphere" to be clear, since the mentioned satellite measurements of BrO (GOME) primarily provide the the total column BrO.

3) Page 2 line 8: Due to the large variations in stratospheric column BrO in high latitudes, polar tropospheric ozone depletion can be identified by satellite measurement only if stratospheric BrO loading is properly addressed, which should be mentioned here.

4) Page 6 line 8: "The stratospheric BrO distribution varies little with latitude and even less with longitude (Theys et al., 2009b)." It is the case only for the low latitude regions; stratospheric column BrO has large longitudinal and latitudinal variations in middle and high latitude regions.

5) Sect 4.3: Choi et al. (2012) have reported the high BrO abundances associated with high planetary boundary layer height in the Arctic region, which can be a relevant reference here.

---

## Referee Comment (RC2) · Anonymous Referee #2 · 18 May 2016

This manuscript describes measurements of BrO above the Rann of Kutch in India/Pakistan from satellite remote sensing. The work is interesting and well presented, although some use of English should be improved. The scope of the work fits in the journal.

General comments:

The manuscript describes potential albedo effects on the retrieval of tropospheric BrO and tries to argue that albedo is not a contributor to the enhanced BrO observed in this region. That discussion needs further consideration. It is clear from the albedo maps (Fig. 4) that the "white desert" of the Rann of Kutch is higher albedo than surroundings, or during Monsoon is similar to surroundings. The text description indicates

that the albedo is lower during the time of the April/May peak, which is true in the sense that of the absolute albedo in the Rann compared to other times of year, but the albedo contrast between the Rann and surrounding areas appears largest during the Feb-May time period. Significantly, the albedo contrast between the Rann and surroundings during the monsoon is very small, so if albedo affects BrO retrieval, no differential albedo exists during the monsoon and no BrO enhancement would be expected. The method of removing a background surrounding the Rann (Eqn. 1 on page 6) is potentially sensitive to the differential albedo between the Rann region and the surroundings. Therefore, I would suggest plotting on Fig. 5 not the reflectivity in the Rann region, but instead the difference in reflectivity between the Rann and the "background" region used for removal of stratospheric BrO influence. At least by eye, this seems to have a pattern more like the BrO enhancement. However, it does appear that the winter season is different than springtime despite similar albedo difference (Rann minus surrounding background regions). That seasonal difference could be affected by stratospheric annual cycling and should be further considered.

The discussion of GOME-2 data and comparison to OMI is less well developed than other aspects of the work. A number of arguments are made, but none are really fully explored. For example, there is a discussion of the diurnal cycle of BrO that is indicated to potentially be the cause of lower BrO abundance at the time of GOME-2 overpass (morning) compared to OMI (early afternoon). At most clean polar sites, BrO cycles are not highly diurnally varying, which is due to production of Br2 in the prior evening and at night and rapid photolysis of this brown gas in the early morning. Therefore, the supposed cycle at least would differ from polar sites. A reference, Holla et al. (2015), is cited, which does indeed show a peaking of BrO later afternoon. However, the Holla et al. (2015) manuscript also shows NO2 data that are enhanced through trapping of pollution NO2 in the shallow nocturnal / early morning boundary layer. Levels of NO2 above $\sim$1 nmol/mol appear to prevent production of high BrO levels. Therefore, the dirunal cycle at the Dead Sea may not be appropriate to the Raan of Kutch. In fact, the manuscript doesn't consider regional pollution, when it could have an effect on these

data. Nearby Karachi has a population of ∼24 million people, and Ahmadabad is ∼7 million. Other effects like boundary layer height, morning fog/clouds, etc. should be fully discussed and this section should be revised accordingly.

The boundary layer height is not treated consistently in this manuscript. In the section about GOME-2 data, 2km (GOME-2 overpass) and 3km (OMI overpass time) are quoted, but the Table 1 and AMF calculations appear to use a 1km ABL height, with alternatives of 0.4 and 2.0 km. This is not consistent and clearly from the sensitivity testing in Table 1, would change the peak BrO mixing ratio significantly.

The manuscript should indicate that this work is motivation for measuring BrO from the ground in this region to verify the space-borne observations. Although the region is clearly remote, it is not inaccessible, nor would measurements via simple MAX-DOAS systems be difficult. Verifying the presence or absence of this space-detected feature would significantly contribute to our ability to connect space-based measurements to ground truth.

A number of sections discuss that the Rann of Kutch "...is probably one of the strongest natural point sources of reactive bromine...", but there is no comparison of this source to other sources. Additionally, the manuscript indicates that there is "supposed to have significant impact on local and regional ozone chemistry", but there is no calculation showing that this impact is significant. It certainly "may" have an impact, but in the absence of some reference indicating significance, the wording appears inaccurate.

Specific comments:

page 1, line 11: indicate that the times are "respectively"

page 1, line 13: replace "former" with "prior"

page 1, line 15: reword "supposed to have a significant influence"

page 1, line 19: Missing "von" from "von Glasow"

page 1, line 21: move "significantly" to after "troposphere"

page 2, line 19: cut the word "ever"

page 2, line 27: are both BrO and IO below 2 ppt? clarify

page 4, BrO retrieval: This section appears to indicate that Level 1 OMI data were re-analyzed by this group rather than use of the OMI BrO product (OMBRO). Can this be made more clear, and the specific sources of the data from OMI data streams should be described fully. If this calculation differs significantly from OMBRO, that should be noted – why was OMBRO not used?

page 6, line 15: This section is not fully clear. Why are slant columns of BrO (S*_trop) being calculated? Which "geometrical AMF"? Doesn't a geometrical AMF assume that the reflector is the Earth's surface, while the actual tropospheric return may be from clouds / fogs / aerosol light scattering?

page 6, Radiative transfer section. This seems like it should use at least 2km layer

page 6, line 28: "...and briefly described in Carn..."

page 7, line 25: Effects of local pollution may be affected by wind direction

page 8, line 18: should say "...not a strong..."

page 12, after line 16: should discuss potential of morning fog and/or NO2

page 13, line 1,2: boundary layer height inconsistent with modeling.

page 13, line 28: why is Salar de Uyuni being discussed here?

page 14, line 11: clarify "supposed to"

page 15, line 14: Wasn't OMAEROG also used, as well as either some level 1 OMI data or OMBRO (unclear).

page 22, caption says "adapted from Fund, W."? Typo?

---

## Author Comment (AC1) · 12 Aug 2016

We would like to thank Referee #1 for the detailed and helpful comments and suggestions he/she made to improve the quality and clarity of our manuscript.
For reference, the original comments (**black**) are always included below, followed by our response (**blue**). Modifications of the original manuscript (**green**) are indicated in **red**.

Hörmann et al. report seasonal variations of tropospheric BrO abundances over the Rann of Kutch (salt desert/seasonal lake at the border of India and Pakistan) using BrO tropospheric column retrieved using OMI UV measurement during 2005-2014. This is a first attempt to quantify tropospheric BrO over salt lakes using satellite measurements. This study agrees well with the scope of Atmospheric Chemistry and Physics. I recommend this article to be published in ACP given that the following major and minor concerns are addressed.

Major comments:

1) Total column BrO retrieval using DOAS (Differential Optical Absorption Spectroscopy) has significant uncertainty depending on selection of fitting window, up to ≈50% or higher, which may affect the magnitude of tropospheric column BrO and thus the BrO mass abundance quantified using that column. The tropospheric column BrO and BrO mass abundance is as correct as the magnitude of total column BrO retrieved in the given fitting window, and it should be mentioned in the manuscript (in Sect. 3). In addition, comparing the BrO mass abundances from authors' retrieval and those from the OMBRO operational product (using the same approach) will give a good, solid example to show possible uncertainties in mass abundances calculated from satellite BrO measurements.

We agree with the referee that the uncertainty of the retrieved slant column density depends on the selection of the fit window as thoroughly discussed by Vogel et al. (2015). In addition, the final total (and especially the tropospheric) column density/BrO mass depends on the a-priori assumptions used for the radiative transfer calculations.
We compared the results of our own OMI BrO retrieval at MPIC with those from the NASA OMBRO operational product in order to illustrate the differences as suggested by the referee. The calculation of tropospheric BrO VCDs was therefore applied to the BrO slant columns from the OMBRO product in the very same way as for the SCDs retrieved at MPIC. Mean BrO VCDs were calculated for April (the month with the highest BrO VCDs) and October (close to zero BrO VCDs directly after the monsoon season) for the whole time period 2005-2014. Finally, the resulting BrO VCDs over the Rann area (22.5-25.5°N, 67.5-72.5°E) were integrated along the longitudinal direction (yielding so called *line densities*) to allow an easy comparison of the results (Figure R1).

[Figure]

[Figure]

Fig. R1: Comparison of the BrO line densities from the NASA OMBRO product (red) with our (MPIC) evaluation (blue) for April (top panel) and October (bottom panel) of the 2005-2014 period.

As can be seen from Figure R1, the resulting BrO line densities for the MPIC and NASA data are very similar, showing essentially the same features and differences of typically 10%.
We added this information to the manuscript at the end of the first paragraph of the "*Results*" section:

"A detailed comparison of the BrO VCDs with those from the operational NASA OMBRO product showed only small differences of typically 10%. For this purpose, the calculation of troposphere BrO VCDs was applied to the BrO slant columns from the OMBRO product in the very same way as for the SCDs retrieved at MPIC."

**Additional reference**:

Vogel, L., Sihler, H., Lampel, J., Wagner, T., and Platt, U.: *Retrieval interval mapping: a tool to visualize the impact of the spectral retrieval range on differential optical absorption spectroscopy evaluations*, Atmos. Meas. Tech., 6, 275-299, doi:10.5194/amt-6-275-2013, 2013.

2) Although stratospheric column BrO has little variations at the region of study, the "magnitude" of assumed stratospheric column BrO can affect the magnitude of tropospheric column BrO and thus the magnitude of mass abundances of tropospheric BrO. Such uncertainties in the magnitude are particularly important in this study, since this study aims "quantification" of BrO mass abundances instead of merely tracking the seasonal and temporal variations. How does the estimate of stratospheric column BrO compare with the other estimates using models, such as Theys et al. 2009b and Salawitch et al., 2011? How much can the estimated tropospheric BrO abundances change along with the assumed loadings of stratospheric column BrO? These points need to be addressed in the manuscript (probably in Sect. 3.2).

We agree with the reviewer that a good estimation of the stratospheric BrO column is a prerequisite for the correct quantification of the tropospheric BrO VCD. However, as the enhanced BrO columns over the salt marsh can be expected to be tropospheric (e.g. because of the spatial correlation to the Rann surface), we applied a simple approach to remove the stratospheric BrO contribution by the empirical background correction described in Section 3.2. The advantage of this empirical correction is that it depends on few parameters for the estimation of the stratospheric BrO column and, thus, interferences are effectively avoided. Compared to the spatial extent of the Rann area, stratospheric BrO gradients over the studied area are comparatively small so that they can be modelled/fitted by a simple 2-dimensional (spatial) polynomial.

In order to make this point clearer in the manuscript, we added the following sentences to Section 3.2: "Please note that this rather simple approach can only be applied, because the stratospheric BrO gradients over the studied area are relatively small compared to the extent of the salt marsh. In general, an accurate quantification of tropospheric BrO VCDs (e.g. at high latitudes during arctic spring) needs a more sophisticated estimation of the stratospheric BrO contribution (e.g. Sihler et al., 2012)."

3) Sect 4.5: Authors used a geometrical AMF for tropospheric BrO over the Kutch of Rann using GOME-2 data. However, I do not agree with that the BrO column retrieved using nadir-viewing UV measurements and a geometrical AMF over not-so-bright surfaces (albedo ≈ 0.15) has the capability to distinguish tropospheric BrO contribution. If authors cannot prove that using a geometrical AMF has such capability or provide the GOME-2 BrO analysis using tropospheric AMF from reasonable radiative transfer calculation, the entire section need to be removed.

We agree in principle with the referee that a geometrical AMF is not well suited to quantitatively determine the BrO VCD over areas with a relatively low albedo, as a tropospheric BrO enhancement may be underestimated. However, in this subsection, the aim was to perform a qualitative comparison, since for the GOME-2 measurements, the correct calculation of a AMF is hindered by the fact that the GOME-2 cloud product is not

reliable (due to the high surface reflectivity). Also, as mentioned in the beginning of Section 4.5 of the manuscript, the GOME-2 cloud product cannot be used for sorting out measurements that were affected by clouds without losing a significant amount of measurements over the cloud free salt marsh (see also Section 3). Therefore we decided to use geometric AMFs for both satellite instruments in order to ensure a qualitative seasonal intercomparison.

To make this point more clear, we added the following sentence to Section 4.5 of the revised manuscript: "Although geometrical AMFs are not well suited to quantitatively determine the BrO VCD over the Rann, they still can be used to visualize BrO abundances exceeding the comparably smooth stratospheric background."

Minor comments:

1) Latitude and longitude of the Rann of Kutch need to be specified in the early part of the manuscript for readers who are not familiar with the area of study.

   We added coordinates at the beginning of Section 1.2 ("...about 22.5-25.5°N, 67.5-72.5°E")

2) Page 2 line 6: "an overall picture of the BrO horizontal distribution" need to be "an overall picture of BrO horizontal distribution in the troposphere" to be clear, since the mentioned satellite measurements of BrO (GOME) primarily provide the total column BrO.

   We changed the sentence to:
   "In the meantime, an overall picture of the total BrO horizontal distribution was obtained by observations from satellite instruments in the late 1990s (Wagner and Platt, 1998; Richter et al., 1998; Chance, 1998)."
   As the referee correctly mentions, the satellite measurements provided just the total column of BrO in the first place.

3) Page 2 line 8: Due to the large variations in stratospheric column BrO in high latitudes, polar tropospheric ozone depletion can be identified by satellite measurement only if stratospheric BrO loading is properly addressed, which should be mentioned here.

   We changed the sentence to read as follows: "A detailed analysis strongly indicated that the ODEs can be associated with huge tropospheric BrO 'clouds' of several thousands of km² extent, '(…) rather than by a disturbance of the stratospheric composition or a modification of the stratospheric AMF' (Wagner and Platt, 1998)."

4) Page 6 line 8: "The stratospheric BrO distribution varies little with latitude and even less with longitude (Theys et al., 2009b)." It is the case only for the low latitude regions; stratospheric column BrO has large longitudinal and latitudinal variations in middle and high latitude regions.

   We changed the sentence to read as follows: "The stratospheric BrO distribution for lower latitude regions (like the Rann of Kutch area) varies little with latitude and even less with longitude (Theys et al., 2009b)."

Sect 4.3: Choi et al. (2012) have reported the high BrO abundances associated with high planetary boundary layer height in the Arctic region, which can be a relevant reference here.

We changed the beginning of Section 4.3. as follows: "Several studies (e.g. Theys et al., 2009b, Choi et al., 2012, Sihler et al., 2012) discussed the influence of a high boundary layer height on the BrO satellite retrieval over Arctic regions. Recently, Lieb-Lappen and Obbard (2015) analysed the role of blowing snow in the activation of bromine over first-year Antarctic sea ice."

---

## Author Comment (AC2) · 12 Aug 2016

We would like to thank Referee #2 for the detailed and helpful comments and suggestions he/she made to improve the quality and clarity of our manuscript.
For reference, the original comments (**black**) are always included below, followed by our response (**blue**). Modifications of the original manuscript (**green**) are indicated in **red**.

This manuscript describes measurements of BrO above the Rann of Kutch in India/Pakistan from satellite remote sensing. The work is interesting and well presented, although some use of English should be improved. The scope of the work fits in the journal.

General comments:

The manuscript describes potential albedo effects on the retrieval of tropospheric BrO and tries to argue that albedo is not a contributor to the enhanced BrO observed in this region. That discussion needs further consideration. It is clear from the albedo maps (Fig. 4) that the "white desert" of the Rann of Kutch is higher albedo than surroundings, or during Monsoon is similar to surroundings. The text description indicates that the albedo is lower during the time of the April/May peak, which is true in the sense that of the absolute albedo in the Rann compared to other times of year, but the albedo contrast between the Rann and surrounding areas appears largest during the Feb-May time period. Significantly, the albedo contrast between the Rann and surroundings during the monsoon is very small, so if albedo affects BrO retrieval, no differential albedo exists during the monsoon and no BrO enhancement would be expected. The method of removing a background surrounding the Rann (Eqn. 1 on page 6) is potentially sensitive to the differential albedo between the Rann region and the surroundings. Therefore, I would suggest plotting on Fig. 5 not the reflectivity in the Rann region, but instead the difference in reflectivity between the Rann and the "background" region used for removal of stratospheric BrO influence. At least by eye, this seems to have a pattern more like the BrO enhancement. However, it does appear that the winter season is different than springtime despite similar albedo difference (Rann minus surrounding background regions). That seasonal difference could be affected by stratospheric annual cycling and should be further considered.

The albedo indeed contributes to the BrO enhancement observed over the Rann area and is therefore considered for the radiative transfer calculations (Section 3.3). Nevertheless, the BrO distribution cannot be generally explained by the influence of the bright Rann surface as there is no clear correlation between the observed reflectivity and the BrO VCDs.
The referee correctly notes that the contrast between the Rann and the surroundings changes during the year, not only because of the varying absolute albedo of the salt marsh, but also because of that in the surrounding region. It is therefore very important to make sure that the "differential albedo" does not affect the polynomial stratospheric background correction and (in a worst case scenario) might lead to a spurious enhancement/decrement of the final retrieved BrO VCDs over the Rann.
To minimize this risk, an extensive area surrounding the Rann is used for the stratospheric background correction (18-30°N and 62-78°E; this essentially encompasses the whole area shown in the monthly BrO maps in Figure 3). The actual Rann area (22.5-25.5°N and 67.5-72.5°E) is completely

excluded from the 2D polynomial fit to make sure that the differential albedo between the Rann and the surrounding area doesn't affect this approach.

Following the suggestion of Referee #2 and to further demonstrate that the albedo contrast can not explain the BrO patterns, we normalized the reflectivity of the Rann region by the mean reflectivity in a nearby area (the reference area mentioned in Section 4.2: 22.5-25.5°N/62-67°E) for all months during the time period 2005-2014. To illustrate the change of contrast between Rann and surrounding areas, Figure X1 (right) shows the normalized reflectivity in longitudinal direction over the Great Rann (24°N latitude) along with the increasing BrO VCDs (left) during March-May 2005-2014 (colour coded in blue – March, red – April and black – May). While the BrO VCDs are already enhanced during March, peak in April and clearly decrease and shift towards East in May, it is still obvious from the normalized reflectivity that there is a continuous decrease of the Rann albedo at the same time (please note that the enhanced reflectivity at 67.5°E during May results from increasing cloud coverage at the western coast in the run-up to the monsoon season).

[Figure]

Fig X1: Mean BrO VCD (left panel) and differential reflectivity (right panel) in longitudinal direction over the Great Rann of Kutch at 24°N latitude for March-May (colour coded) 2005-2014.

During winter time (December-February), the differential reflectivity over the Rann (and therefore the contrast between the Indian Ocean and the salt marsh) is even higher, but shows only little variation (Fig. X2, right panel). The corresponding BrO VCDs are only slightly enhanced and also show very small variations, although the upcoming BrO increase seems to already start slowly in February (Fig. X2, left panel).

[Figure]

Fig. X2: Mean BrO VCD (left) and differential reflectivity (right) in longitudinal direction over the Great Rann of Kutch at 24°N latitude during December-February 2005-2014 (colour coded in blue – December, red – January and black – February).

Another important finding is that the spatial patterns of the reflectivity and the BrO VCDs are quite different, indicating that the observed enhanced tropospheric BrO VCDs are not a possible artefact caused by the albedo contrast.

We included the results of this accompanying study as an additional section in the Supplementary Material and added the following reference to Section 4.1 of the manuscript:
"While there is only little variation in the regional reflectivity within the Rann area over different months, the spatial distribution of BrO VCDs changes and even seems to progress into the eastern part of the Great Rann from March-May (see Supplementary Material for a more detailed investigation of these effects)."

The discussion of GOME-2 data and comparison to OMI is less well developed than other aspects of the work. A number of arguments are made, but none are really fully explored. For example, there is a discussion of the diurnal cycle of BrO that is indicated to potentially be the cause of lower BrO abundance at the time of GOME-2 overpass (morning) compared to OMI (early afternoon).

We agree with the referee that the discussion of this point is rather qualitative. The reason for this is that the GOME-2 cloud product is not very reliable over the bright surface of the Rann. However, the aim of this study was a relative comparison of both data sets (after they were processed in the same way). We made this more clear in the revised version of the manuscript (see also answer to Referee#1).

At most clean polar sites, BrO cycles are not highly diurnally varying, which is due to production of $Br_2$ in the prior evening and at night and rapid photolysis of this brown gas in the early morning. Therefore, the supposed cycle at least would differ from polar sites. A reference, Holla et al. (2015), is cited, which does indeed show a peaking of BrO later afternoon. However, the Holla et al. (2015) manuscript also shows $NO_2$ data that are enhanced through trapping of pollution $NO_2$ in the shallow nocturnal / early morning boundary layer. Levels of $NO_2$ above ≈1 nmol/mol appear to prevent production of high BrO levels. Therefore, the diurnal cycle at the Dead Sea may not be appropriate to the Rann of Kutch. In fact, the manuscript doesn't consider regional pollution, when it could have an effect on these data. Nearby Karachi has a population of ≈24 million people, and Ahmadabad is ≈7 million.

We like to thank the reviewer for the valuable note! It is true that the supposed BrO cycle at the Dead Sea in Holla et al. (2015) is often influenced by enhanced levels of $NO_2$ and therefore differs from those observed in clean polar sites and probably also at the Rann of Kutch. For the Dead Sea, Holla et al. (2015) found mutually exclusive enhanced BrO and $NO_2$ abundances in time and space, suggesting a conversion of $NO_2$ into $BrONO_2$ and therefore suppressing the *bromine explosion* during the early morning. This hypothesis was additionally supported by typically observed diurnal variations of the synoptic winds and meteorology.
In contrast to the Dead Sea area, satellite observations by both considered satellite instruments (OMI and GOME-2) over the Rann of Kutch only show a very localized $NO_2$ enhancement close to the cities of Karachi (≈ 300-400 km away) and Ahmadabad (≈ 200-300 km away) due to the short $NO_2$ lifetime of several hours. Close to the salt marsh, the observed tropospheric $NO_2$ VCDs are typically close to background levels. A possible influence of $NO_2$ at the Rann of Kutch can therefore expected to be much less important than over the Dead Sea basin. For corresponding tropospheric $NO_2$ maps for OMI and GOME-2, please visit e.g. the TEMIS website (available via http://www.temis.nl/airpollution/no2col/no2regioomimonth_v2.php).

We revised argument #3 of Section 4.5 as follows:

"3. One of the main chemical reasons for much lower BrO VCDs using GOME-2 might be that the measurements take place about 4 hours earlier when compared to OMI observations (≈ 9:30 vs. 13:30 LT). At the time of the morning overpass, the 'bromine explosion' mechanism has presumably not progressed very far, as solar irradiance is approximately 50% less than during OMI's afternoon overpass (according to ECMWF data) and the process is photolytically driven. Furthermore, $O_3$ is needed for the rapid build-up of BrO, which might be more easily available during the morning on the one hand, but may lead to differences in the spatial BrO distribution patterns for GOME-2 when compared to OMI. Observations at the Dead Sea (Israel) have shown that the largest BrO VCDs may show up close to noontime, if enhanced $NO_x$ levels are generally present due to anthropogenic pollution (Holla et al., 2015). In contrast to the Dead Sea area, however, neither OMI nor GOME-2 observations show significantly enhanced $NO_2$ VCDs over the Rann of Kutch area, but only in the vicinity of the cities Karachi (≈ 300-400 km away) and Ahmadabad (≈ 200-300 km away). A possible influence of $NO_2$ at the Rann of Kutch is therefore assumed to be much less important than over the Dead Sea."

Other effects like boundary layer height, morning fog/clouds, etc. should be fully discussed and this section should be revised accordingly.

The possible effects of the BLH are briefly discussed on p.13, argument #4, while an influence of clouds (and/or morning fog) due to the different overflight time of the instruments could not be explained from MODIS and ECMWF data (as mentioned in Section 4.5). Furthermore, we are convinced that the best way to conduct such an analysis would be an extensive ground-based local measurement campaign, where the environmental parameters may be observed with much better temporal resolution during the day at the same time as the BrO (and possibly $NO_2$) column.
To emphasize this, we modified the end of Section 4.5 as follows:
"Although all of these effects probably contribute to the observed differences between OMI and GOME-2 observations, the reasons for these discrepancies are still a matter of further research. The best way to analyze the influence of ambient conditions on BrO formation would be an extensive ground-based measurement campaign. Such local observations would provide high temporal resolution data of meteorological parameters like wind speed and direction, relative humidity, boundary layer height, clouds (and morning fog) as well as a possible influence of anthropogenic pollution."

The boundary layer height is not treated consistently in this manuscript. In the section about GOME-2 data, 2km (GOME-2 overpass) and 3km (OMI overpass time) are quoted, but the Table 1 and AMF calculations appear to use a 1km ABL height, with alternatives of 0.4 and 2.0 km. This is not consistent and clearly from the sensitivity testing in Table 1, would change the peak BrO mixing ratio significantly.

The referee is right that (according to the ECMWF data) the boundary layer height during the overflight times of the satellite instruments is 2 and 3 km (GOME-2 and OMI), respectively, which may lead to differences of the BrO layer profile and therefore the true BrO VCD. However, the BrO VCDs in Section 4.5 are only calculated by using a geometrical AMF for both instruments (GOME-2 and OMI), which means that neither a specific BrO layer profile nor the boundary layer height is taken into account for the AMF calculation. A more sophisticated AMF calculation for the GOME-2/OMI section remains difficult due to problems with the GOME-2 operational cloud product (see former discussion), so we decided to only show a qualitative comparison of the data. However, a different treatment of GOME-2 and OMI data by assuming BrO layer heights of 2km respectively 3km would result in only about 15 % differences for the resulting BrO VCDs and cannot explain the GOME-2 BrO VCDs close to background level.
It is furthermore important to note that while it can be expected that a change of the boundary layer height may lead to changes of the BrO layer profile, the true BrO profile over the Rann remains unknown (the BrO layer doesn't necessarily need to fit the boundary layer height). The presumed

baseline scenario (homogeneous BrO layer between 0-1km) only represents a first guess of the true BrO profile, based on the assumption that BrO forms at the salt surface and is partly transported to higher altitudes. To indicate the large uncertainties of the derived BrO mixing ratios, Table 1 shows results for two other a priori BrO profiles. However, for an adequate estimation of the true BrO profile, results from local ground-based MAX-DOAS measurements are desirable.

To emphasize the assumptions made, we modified the manuscript as follows:

Section 3.3 (end): "It is furthermore important to note that the baseline scenario (assuming a homogeneous BrO layer between 0-1km) only represents a first guess of the true BrO profile, based on the assumption that BrO forms at the salt surface and is partly transported to higher altitudes. While the BrO profile can be expected to depend on the prevailing boundary layer height, the true profile over the Rann remains unknown (the BrO layer doesn't necessarily need to fill in the complete boundary layer). An estimate of the uncertainties caused by the a priori assumptions will be given in Section 4.2. For an adequate estimation of the true BrO profile, however, results from local ground-based MAX-DOAS measurements would be desirable in the future."

Section 4.5 (argument #4): "4. The boundary layer height during the GOME-2 overflight in April/May is significantly lower ($\approx$ 2km) than for the OMI measurements ($\approx$ 3km). As the BLH increases towards noon, BrO originating from the ground might be transported to higher altitudes where it could be more easily detected by the OMI as the instrument's sensitivity generally increases for elevated layers. Additionally, the increasing BLH might lead to an increased mixing-in of tropospheric $O_3$ from higher altitudes and thereby lead to further formation of BrO. It is important to note that the data shown in Figure 4 were only calculated by application of a geometrical AMF and therefore possible differences for the BLH and BrO profile are not taken into account. However, in the case of a homogeneous BrO layer filling in the complete boundary layer, corresponding radiative transfer effects would only result in about 15% differences for the resulting BrO VCDs and can therefore not explain the GOME-2 BrO VCDs close to background level."

The manuscript should indicate that this work is motivation for measuring BrO from the ground in this region to verify the space-borne observations. Although the region is clearly remote, it is not inaccessible, nor would measurements via simple MAX-DOAS systems be difficult. Verifying the presence or absence of this space-detected feature would significantly contribute to our ability to connect space-based measurements to ground truth.

We totally agree with the referee and modified the manuscript as specified in our answers to the previous comments.

A number of sections discuss that the Rann of Kutch "...is probably one of the strongest natural point sources of reactive bromine...", but there is no comparison of this source to other sources. Additionally, the manuscript indicates that there is "supposed to have significant impact on local and regional ozone chemistry", but there is no calculation showing that this impact is significant. It certainly "may" have an impact, but in the absence of some reference indicating significance, the wording appears inaccurate.

We agree with the referee that it is important to emphasize that there are still only few measurements of halogen formation over salt lakes worldwide. To our knowledge, these are the first reported observations of BrO over a salt lake (salt marsh) by a satellite instrument at all. However, there are at least two main points, why the Rann of Kutch is "…probably one of the strongest natural point sources of reactive bromine…" and halogen emissions from salt lakes may "…have significant impact on local and regional ozone chemistry":

1. By looking at global BrO maps using OMI data, the Rann of Kutch appears to be the only salt lake/marsh where a clear enhancement of the BrO column can be easily seen during several months from satellite measurements, even without correcting for the stratospheric background and excluding clouded data.

2. Although the actual BrO profile over the Rann of Kutch is unknown, a qualitative comparison to the BrO VCDs over the Dead Sea indicates that the BrO abundance over the Rann of Kutch is much larger than over the Dead Sea. So far, the largest BrO mixing ratios at a salt lake were reported from ground-based DOAS measurements at the Dead Sea (up to 220 ppt, as reported in Matveev et al., 2001 and Tas et al., 2005).
It is nevertheless likely that the BrO VCDs over the Rann of Kutch (as well as the Dead Sea) are generally biased low because of spectral dilution due to the low spatial resolution of satellite instruments like OMI when compared to local ground-based measurements (compare to argument #1 of the GOME-2 discussion on p.13 of the revised manuscript).

Saiz-Lopez and von Glasow (2012) state that about 2.5% of the global land surface *"(…) is covered by saline soils, implying that halogen release might be relevant on a rather large part of the continents and not only over the comparatively small areas of salt lakes"*. However, they also noted that *"A regional or global assessment of the relevance of halogen chemistry over salt lakes and saline soils is so far missing"*. To further classify the significance of the BrO abundance in comparison to other salt lakes, local ground-based MAX-DOAS measurements are highly desirable (as mentioned several times before).

We added a short paragraph to Section 4.6 of the manuscript to strengthen the given statements as follows:

"It is interesting to note that the BrO VCDs found over the Dead Sea are generally much lower than those observed over the Rann of Kutch, although the above-mentioned ground-based measurements showed the highest BrO mixing ratios so far observed at a salt lake (up to 220 ppt). This finding indicates that the Rann of Kutch is likely one of the strongest natural point sources of reactive bromine compounds outside the polar regions. This argument is further strengthened by the fact that the Rann is the only salt lake/marsh, where a clear enhancement of the BrO column can be easily seen during several months from satellite measurements (even without correcting for the stratospheric background or excluding clouded data). However, BrO from other salt lakes (particularly smaller ones, like the Dead Sea) may generally not be identified as easily from the OMI data."

Specific comments:

page 1, line 11: indicate that the times are "respectively" done

page 1, line 13: replace "former" with "prior" done

page 1, line 15: reword "supposed to have a significant influence"

We reformulated the sentence: *"The measurements indicate that the Rann of Kutch salt marsh is one of the strongest natural point sources of reactive bromine compounds outside the polar regions."*

page 1, line 19: Missing "von" from "von Glasow" done

page 1, line 21: move "significantly" to after "troposphere" done

page 2, line 19: cut the word "ever" done

page 2, line 27: are both BrO and IO below 2 ppt? clarify

We reformulated the sentence as follows: "However, none of these measurements showed significantly enhanced BrO or iodine oxide (both less than 2 ppt or below the detection limit of the MAX-DOAS measurements)."

page 4, BrO retrieval: This section appears to indicate that Level 1 OMI data were reanalysed by this group rather than use of the OMI BrO product (OMBRO). Can this be made more clear, and the specific sources of the data from OMI data streams should be described fully. If this calculation differs significantly from OMBRO, that should be noted – why was OMBRO not used?

Our group has a long-standing experience in spectral retrievals of satellite observations, and was one of the first to analyse BrO from satellite. Thus in this study we applied our own algorithm for the analysis of the satellite data (OMI and GOME-2). As shown in the reply to Referee #1, almost the same results are obtained if the OMBRO product is used. We added this information to the text:

The spectral data of both satellite instruments were analysed at MPI-C for BrO column densities by using the Differential Optical Absorption Spectroscopy (DOAS) technique....
The OMI BrO retrieval follows the general settings of the GOME-2 retrieval: Level 1 data (online provided by the Goddard Earth Sciences Data & Information Services center, NASA) were analysed using a wavelength range from 336--360 nm, including 4 adjacent BrO absorption bands.

page 6, line 15: This section is not fully clear. Why are slant columns of BrO (S*_trop) being calculated? Which "geometrical AMF"? Doesn't a geometrical AMF assume that the reflector is the Earth's surface, while the actual tropospheric return may be from clouds / fogs / aerosol light scattering?

A geometrical AMF indeed assumes that the detected sunlight directly traversed the complete atmosphere and is reflected by the Earth's surface alone. However, if the main absorber can be assumed to be mainly present in the stratosphere, the geometrical AMF is a viable approximation for the actual AMF. In the manuscript, the geometrical AMF is solely used for an initial calculation of the main effect of the satellite's measurement geometry and the stratospheric BrO background correction. Please additionally see our answer to Referee #1.
We made this point now more clear in the manuscript.

page 6, Radiative transfer section. This seems like it should use at least 2km layer

Please see comments above

page 6, line 28: "...and briefly described in Carn..." done

page 7, line 25: Effects of local pollution may be affected by wind direction

Please see comments above

page 8, line 18: should say "...not a strong..." done

page 12, after line 16: should discuss potential of morning fog and/or NO2

Please see comments above

page 13, line 1,2: boundary layer height inconsistent with modeling.

Please see comments above

page 13, line 28: why is Salar de Uyuni being discussed here?

Salar de Uyuni is the largest salt flat in the world and clearly enhanced BrO column densities have been detected by ground-based DOAS measurements in the past. It is briefly mentioned to make clear why other large salt lakes/marshes may not be easily observed by satellites using the DOAS technique due to spectral effects over high and bright surfaces.
We added this information to the manuscript.

page 14, line 11: clarify "supposed to"

Please see comments above

page 15, line 14: Wasn't OMAEROG also used, as well as either some level 1 OMI data or OMBRO (unclear).

We added the missing information to the Acknowledgement

page 22, caption says "adapted from Fund, W."? Typo?

Changed to "adapted from World Wildlife Fund:…"

---

## Author Response (AR2)

Submitted on 08 Sep 2016

We would like to thank Referee #1 once again for the suggestions he/she made to improve the quality and clarity of our manuscript.
For reference, the original comments (**black**) are included below, followed by our response (**blue**).
Modifications of the original manuscript (**green**) are indicated in **red**.

Major comment 2:

The question, "How does the estimate of stratospheric column BrO compare with the other estimates using models, such as Theys et al. 2009b and Salawitch et al., 2011? How much can the estimated tropospheric BrO abundances change along with the assumed loadings of stratospheric column BrO?" has not been answered in the revised manuscript.

I suggest authors to state the difference in magnitude between their stratospheric column and the model study and give the estimate of the difference in tropospheric column that may caused by employing different stratospheric column, even in a broad range.

The application of our empirical 2D-polynomial stratospheric BrO model yields only slightly varying monthly averaged BrO VCDs in the range of $2.5 - 3 \times 10^{13}$ molec/cm² above the considered Rann of Kutch area. To further illustrate the approach, the resulting monthly averaged stratospheric BrO VCDs maps for 2005 are presented in Figure Z1.

A significant influence of the stratospheric BrO estimation on the finally retrieved tropospheric BrO columns over the Rann of Kutch area can nevertheless be excluded. The monthly averaged variation of stratospheric BrO takes place at much larger spatial scales compared to the variations of the BrO enhancement observed over the Rann of Kutch (please note that the colorbar in Figure Z1 has been scaled to show the maximum contrast). Generally, the tropospheric BrO column can be assumed to be close to zero for low/mid latitudes. The area considered in this study encompasses latitudes from 18-30°N. By subtracting the daily 2D spatial polynomial from the total BrO columns, the data for the entire considered area is normalized to (stratospheric) background conditions. To make sure that the enhanced tropospheric BrO columns over the Rann of Kutch have no influence on the stratospheric background correction, the actual Rann area is completely excluded from the polynomial fit and the corresponding stratospheric columns are interpolated by the resulting polynomial as a function of spatial coordinates (as already mentioned in the initial as well as the revised version of the manuscript, Section 3.2).

We don't claim to give an accurate estimation of the stratospheric BrO column on regional scales (as this is not necessary for the presented analysis). Nevertheless, our empirical approach compares well with results from more sophisticated techniques like e.g. the one described in Theys et al. (2009). Figure Z2 shows the monthly zonal averaged stratospheric BrO VCD based on the BASCOE climatology as presented in Theys et al. (2009). The typical stratospheric BrO VCD within the latitudinal band that was considered within our study (18-30°N; indicated by the red rectangle) is about $2.5 \times 10^{13}$ molec/cm². Please note that the data in Figure Z2 is based on global data and calculated for the overpass time of the GOME instrument (around 10:30 LT at the equator).

[Figure]

Figure Z1: Monthly mean estimation of the stratospheric BrO VCD during 2005 as a result from the 2D spatial polynomial approach. The stratospheric BrO variation takes place at much larger spatial scales compared to the variations of the tropospheric BrO enhancement observed over the Rann of Kutch (please compare to Figure 7 in the Supplementary Material).

[Figure]

Figure Z2: Monthly zonal averaged stratospheric BrO VCD during the time of the GOME overpass and based on the BASCOE climatology (adapted version of Figure 13 in Theys et al., 2009). The typical stratospheric BrO VCD within the latitudinal band that was considered within our study (18-30°N; indicated by the red rectangle) is about 2.5 x 10$^{13}$ molec/cm² and shows no significant annual variation.

To improve to quality of the manuscript, we added the following sentences to Section 3.2 of the manuscript: "The stratospheric BrO distribution varies little with latitude, even less with longitude (Theys et al., 2009b) and at much larger spatial scales than the variations of the BrO enhancement observed over the Rann of Kutch. (…) A qualitative comparison of our stratospheric BrO estimation with the more sophisticated approach presented by Theys et al. (2009b) shows similar results and typical $V_{strat}$ of about 2.5 x 10$^{13}$molec/cm$^2$."

Major comment 3:

The authors now use geometric AMF to compare the OMI and GOME-2 BrO abundance in Sect 4.5. However, I am still not convinced with the presented interpretation regarding the difference in the BrO abundances from the two instrument, particularly given the low surface albedo, uncertainty in cloud conditions, and distinct conditions/situations of the two instruments. Overall this article is very lengthy. If the main focus of this article is the seasonal behaves of BrO, removing this section would make the article more cohesive and easier to read.

We respectfully disagree with the referee and remain convinced that neither using a geometrical AMF, differing cloud conditions, surface albedo nor instrumental effects may satisfactorily explain the observed BrO differences for both satellite instruments:

1. **Geometrical AMF**: The reason for applying a geometrical AMF for both instruments was to perform a qualitative seasonal intercomparison, since for the GOME-2 measurements, the correct calculation of an AMF is hindered by the fact that the corresponding cloud product is not reliable. It remains unclear to us why the referee is convinced that using a geometrical AMF could make a substantial difference for the qualitative findings that are presented in Section 4.5 (p. 12, l. 12-14: "…*Although geometrical AMFs are not well suited to*

*quantitatively determine the BrO VCD over the Rann, they still can be used to visualize BrO abundances exceeding the comparably smooth stratospheric background."*). The OMI measurements clearly show that enhanced tropospheric BrO VCDs may be detected over the Rann area, whether geometrical or more sophisticated AMFs are used to convert slant columns into vertical columns.

2. **Surface albedo**: We see no reason why the Rann surface albedo should significantly change within the time period between both satellite observations. Such an effect would indeed make a difference, but appears to be very unlikely.

3. **Influence of cloud conditions**: As we already stated in the revised version of the manuscript (p. 12, l. 17-19), *"…most days outside the monsoon months (June–September) are only effected by low cloud fractions according to MODIS and ECMWF data. In addition, no differences in cloud cover were found that could be attributed to the different overflight time of the instruments throughout the year."*

We regret that the referee is still not convinced with the presented interpretation, but feel confident that our approach is sound and that we have done our very best to responsibly discuss all relevant aspects that might influence observed differences between both satellite instruments.

Furthermore, we think that the results and the corresponding discussion as presented in Section 4.5 ("Comparison of OMI results with GOME-2 observations") are a matter of particular interest. The bromine release/BrO formation process over salt lakes is still largely unknown and being discussed within the scientific community. The differences between the results for both satellite instruments indicate that BrO is likely to be only significantly formed towards noon and the chemical processes are essentially influenced by changing ambient conditions. We therefore think that it is necessary to keep Section 4.5 in the final manuscript.

[revised manuscript text omitted]
) and at much larger spatial scales than the variations of the BrO enhancement observed over the Rann of Kutch. Therefore, BrO VCDs determined assuming a geometrical airmass factor were corrected by subtracting the results of a two-dimensional spatial polynomial fit of 3rd degree $n$ applied to the daily measurements as described in Hörmann et al. (2013):

$$V^*_{\text{trop,i}} \approx V^*_{\text{tot,i}} - \sum_{m,n=0}^{3} V^*_{\text{strat,i}} \times x_i^m \times y_i^n \tag{1}$$

where $V^*_{\text{strat,i}}$ are the fitted stratospheric BrO VCDs at the centre coordinates $x$ and $y$ [°] of satellite pixel $i$ within a large area around the Rann of Kutch (18–30°N, 62–78°E). To minimize the influence of possibly enhanced BrO VCDs over the salt marsh, the actual Rann area (22.5-25.5°N, 67.5–72.5°W) was excluded from the polynomial fit of the stratospheric correction approach. A qualitative comparison of our stratospheric BrO estimation with the more sophisticated approach presented by Theys et al. (2009b) shows similar results and typical $V_{\text{strat}}$ of about $2.5 \times 10^{13}\,\text{molec/cm}^2$. 
[revised manuscript text omitted]

|  |  | 0–2 km | $(27\pm8)$ppt | -23% |
| AOD | 0.7 | $0^*$ | $(32\pm9)$ppt | -8% |
|  |  | 0.4 | $(34\pm10)$ppt | -3% |
|  |  | 1.0 | $(37\pm11)$ppt | +6% |
| surface albedo | 0.15 | 0.1 | $(42\pm12)$ppt | +20% |
|  |  | 0.2 | $(30\pm9)$ppt | -14% |

**Table 2.** Best fits from the multilinear regression analysis in order to model the seasonal BrO mass variation over the Rann of Kutch in dependency of UV surface radiation (UV), precipitation (P), cloud coverage (CC), relative humidity (RH), boundary layer height (BLH) and surface temperature (T).

| # | linear function | $r^2$ |
|---|---|---|
| 1 | $m_{BrO} = a_1 \cdot UV$ | 0.71 |
| 2 | $m_{BrO} = a_1 \cdot UV + a_2 \cdot P$ | 0.83 |
| 3 | $m_{BrO} = a_1 \cdot UV + a_2 \cdot P \qquad + a_4 \cdot RH$ | 0.91 |
| 4 | $m_{BrO} = a_1 \cdot UV \qquad + a_3 \cdot CC + a_4 \cdot RH + a_5 \cdot BLH$ | 0.95 |
| 5 | $m_{BrO} = a_1 \cdot UV + a_2 \cdot P + a_3 \cdot CC + a_4 \cdot RH + a_5 \cdot BLH$ | 0.97 |
| 6 | $m_{BrO} = a_1 \cdot UV + a_2 \cdot P + a_3 \cdot CC + a_4 \cdot RH + a_5 \cdot BLH + a_6 \cdot T$ | 0.97 |